# Descending neuron population dynamics during odor-evoked and spontaneous limb-dependent behaviors

Florian Aymanns, Chin-Lin Chen, Pavan Ramdya*

Neuroengineering Laboratory, Brain Mind Institute & Interfaculty Institute of Bioengineering, EPFL, Lausanne, Switzerland

**Abstract** Deciphering how the brain regulates motor circuits to control complex behaviors is an important, long-standing challenge in neuroscience. In the fly, *Drosophila melanogaster*, this is coordinated by a population of ~ 1100 descending neurons (DNs). Activating only a few DNs is known to be sufficient to drive complex behaviors like walking and grooming. However, what additional role the larger population of DNs plays during natural behaviors remains largely unknown. For example, they may modulate core behavioral commands or comprise parallel pathways that are engaged depending on sensory context. We evaluated these possibilities by recording populations of nearly 100 DNs in individual tethered flies while they generated limb-dependent behaviors, including walking and grooming. We found that the largest fraction of recorded DNs encode walking while fewer are active during head grooming and resting. A large fraction of walk-encoding DNs encode turning and far fewer weakly encode speed. Although odor context does not determine which behavior-encoding DNs are recruited, a few DNs encode odors rather than behaviors. Lastly, we illustrate how one can identify individual neurons from DN population recordings by using their spatial, functional, and morphological properties. These results set the stage for a comprehensive, population-level understanding of how the brain's descending signals regulate complex motor actions.

*For correspondence:
pavan.ramdya@epfl.ch

**Competing interest:** The authors declare that no competing interests exist.

## Editor's evaluation

This article uses a genetically encoded calcium indicator to assess neural activity across a population of axons connecting the fly's brain to its ventral nerve cord while the tethered fly behaves on a floating ball. The preparation and large-scale analysis represent a significant step forward in determining how the brain compresses sensory and state information to convey commands to the ventral nervous system for behavior execution by motor circuits.

## Introduction

The richness of animal behaviors depends on the coordinated actions of many individual neurons within a population. For example, neurons or small networks may compete in a winner-take-all manner to select the next most appropriate motor action (*Cisek and Kalaska, 2010*). To then drive behaviors, the brain conveys these decisions to motor circuits via a population of descending neurons (DNs) projecting to the spinal cord of vertebrates or ventral nerve cord (VNC) of invertebrates. There, DN axons impinge upon local circuits including central pattern generators (CPGs) that transform DN directives into specific limb or body part movements (*Bouvier et al., 2015*; *Capelli et al., 2017*; *Caggiano et al., 2018*; *Orger et al., 2008*). Because DNs make up only about 1% of brain neurons, DN population activity represents a critical information bottleneck: high-dimensional brain dynamics must be

**Figure 1.** Recording descending neuron (DN) population activity and animal behavior. (**a**) Schematic of the *Drosophila* nervous system showing DNs projecting from the brain to motor circuits in the ventral nerve cord (VNC). For clarity, only two pairs of DNs (red and blue) are shown. Indicated (dashed gray line) is the coronal imaging region-of-interest (ROI) in the thoracic cervical connective. (**b**) In a 'modulation' framework for DN population control, new DNs (red) may be recruited to modulate ongoing behaviors primarily driven by core DNs (blue). Each ellipse is an individual DN axon (white, blue, and red). (**c**) In a 'context dependence' framework for DN population control, different DNs may be recruited to drive identical behaviors depending on sensory context. (**d**) Alternatively, in a 'sensory encoding' framework, many DNs may not drive or be active during behaviors but rather transmit raw sensory signals to the VNC. (**e**) An approach for deriving DN cell identity from population recordings. One may first identify sparse transgenic strains labeling specific neurons from DN populations (circled in black) using their functional attributes/encoding, positions within the cervical connective, and the shapes of their axons. Ultimately, one can use sparse morphological data to find corresponding neurons in the brain and VNC connectomes. (**f, g**) Template-registered confocal volume z-projections illustrating a 'brain only' driver line (otd-nls:FLPo; R57C10-GAL4,tub>GAL80>) expressing (**f**) a nuclear (histone-sfGFP) or (**g**) a cytosolic (smGFP) fluorescent reporter. Scale bar is 50 µm. Location of two-photon imaging plane in the thoracic cervical connective is indicated (white dashed lines). Tissues are stained for GFP (green) and neuropil ('nc82', magenta). (**h**) Schematic of the VNC illustrating the coronal (x–z) imaging plane. Dorsal-ventral ('Dor') and anterior–posterior ('Ant') axes are indicated. (**i**) Denoised two-photon image of DN axons passing through the thoracic cervical connective. Scale bar is 10 µm. (**j**) Two-photon imaging data from panel (**i**) following motion correction and $\%\Delta F/F$ color-coding. An ROI (putative DN axon or closely intermingled axons) is indicated (white dashed circle). (**k**) Sample normalized $\%\Delta F/F$ time-series traces for 28 (out of 95 total) ROIs recorded from one animal. Behavioral classification at each time point is indicated below and is color-coded as in panel (**p**). (**l**) Schematic of system for recording behavior and delivering odors during two-photon imaging (not to scale) while a tethered fly walks on a spherical treadmill. (**m**) Spherical treadmill ball rotations (fictive walking trajectories) are captured using the front camera and processed using FicTrac software. Overlaid (cyan) is a sample walking trajectory. (**n**) Video recording of a fly from six camera angles. (**o**) Multiview camera images are processed using DeepFly3D to calculate 2D poses and then triangulated 3D poses. These 3D poses are further processed to obtain joint angles. (**p**) Joint angles are input to a dilated temporal convolutional network (DTCN) to classify behaviors including walking, resting, head (eye and antennal) grooming, front leg rubbing, or posterior (abdominal and hindleg) movements.

*Figure 1 continued on next page*

*Figure 1 continued*

The online version of this article includes the following figure supplement(s) for figure 1:

**Figure supplement 1.** Supporting details regarding neural denoising, driver line expression, odor stimulation, and behavior quantification.

**Figure supplement 2.** The range of joint angles explored during fly behaviors.

compressed into low-dimensional commands that efficiently interface with and are read out by motor circuits. The information carried by individual DNs has long been a topic of interest (*Heinrich, 2002*; *Kien, 1990*; *Böhm and Schildberger, 1992*). Through electrophysiological recordings in large insects, the activities of individual DNs have been linked to behaviors like walking and stridulation (*Heinrich, 2002*). For some DNs, links between firing rate and behavioral features like walking speed have also been established (*Böhm and Schildberger, 1992*). However, how the larger population of DNs coordinate their activities remains unknown.

The fruit fly, *Drosophila melanogaster*, is an excellent model for investigating how DNs regulate behavior. Flies are genetically-tractable, have a rich behavioral repertoire, and have a numerically small and compact nervous system (*Olsen and Wilson, 2008*). *Drosophila* are thought to have between ~350 (*Namiki et al., 2018*) and ~500 (*Hsu and Bhandawat, 2016*) pairs of DNs. Sparse sets of these DNs can be experimentally targeted using transgenic driver lines (*Namiki et al., 2018*) for functional recordings (*von Reyn et al., 2014*; *Schnell et al., 2017*; *Chen et al., 2018*; *Ache et al., 2019b*; *Namiki et al., 2022*) or physiological perturbations (*Cande, 2018*; *Zacarias et al., 2018*; *Ache et al., 2019a*; *Namiki et al., 2022*; *Guo et al., 2022*). The functional properties of DNs can be understood within a circuit context using emerging connectomics datasets (*Zheng et al., 2018*; *Phelps et al., 2021*). Thus, by building upon foundational work in other insects (*Heinrich, 2002*; *Kien, 1990*; *Böhm and Schildberger, 1992*), studies in *Drosophila* can ultimately reveal how identified DNs work collectively to regulate complex behaviors.

Until now, investigations of *Drosophila* have focused on individual or small sets of DNs. These studies have demonstrated that artificial activation of DN pairs is sufficient to drive complex behaviors including escape (*Lima and Miesenböck, 2005*) (giant fiber neurons, 'GF'), antennal grooming (*Hampel et al., 2015*)(antennal descending neurons, 'aDN'), backward walking (*Bidaye et al., 2014*) (moonwalker descending neurons, 'MDN'), forward walking (*Bidaye et al., 2020*) (DNp09), and landing (*Ache et al., 2019a*) (DNp10 and DNp07). These results also suggest a command-like role for some DNs in that they are both necessary and sufficient to drive particular actions (*Kupfermann and Weiss, 1978*).

Although a command-like role for individual DNs is intuitively easy to grasp it may not translate well toward understanding how natural behavior is coordinated by large DN populations. Notably, a behavioral screen revealed that optogenetic activation of most DNs drives only small changes in locomotor and grooming behaviors (*Cande, 2018*) rather than a large variety of distinct actions as might be expected if each DN was a command-like neuron. Therefore, DN populations likely employ additional control approaches. For example, some groups of DNs might modulate or fine-tune actions primarily driven by other command-like DNs. The balance between command-like and modulatory roles of DNs may differ for stereotyped versus flexible behaviors. In line with having a role in behavioral modulation, studies in crickets and locusts have demonstrated that changes in the firing rates of some DNs (*Böhm and Schildberger, 1992*; *Zorović and Hedwig, 2011*) correlate with walking speed and turning (*Heinrich, 2002*). Alternatively, DN subpopulations may represent parallel pathways that are recruited depending on sensory context (*Israel et al., 2022*). For example, different groups of DNs may be differentially engaged during odor-evoked versus spontaneously-generated walking (*Heinrich, 2002*; *Kien, 1990*). Finally, some DNs may convey raw sensory information—instead of be active during behaviors—to enable feedback control of downstream motor circuits. For example, in addition to discovering DNs that are active during steering, one recent study also observed DNa02 activity in response to fictive odors in immobile flies (*Rayshubskiy, 2020*). DNp07 activity has also been observed in response to visual stimulation in nonflying flies (*Ache et al., 2019a*).

To resolve how DNs engage and modulate ongoing behaviors, one would ideally measure the causal relationship between DN activation and behavioral output. Although this has already been done for sparse sets of DNs (*Cande, 2018*), emerging evidence suggests that in many instances DNs act as a population to control behaviors (*Namiki et al., 2022*). Thus, activating specific pairs or groups

of DNs would be extremely valuable. However, achieving the precise co-activation of groups of DNs is technically challenging and combinatorially daunting. One important step is to first identify which DNs are co-active during population recordings in behaving animals. Similarly, to determine the degree to which DNs are recruited depending on sensory context and/or convey raw sensory information from the environment, rather than recording one DN at a time a faster complementary approach would be to record the activity of multiple DNs during the sequential presentation of well-controlled sensory cues. Until now DN population recordings have not been performed due to several technical challenges. First, there has been an absence of tools for selectively genetically targeting DN populations. Additionally, because DN cell bodies and neurites are distributed across the brain (*Namiki et al., 2018*), relatively invasive (*Mann et al., 2017*) volumetric imaging approaches would be required to simultaneously record the activity of many DNs at once. As an alternative, we previously developed a thoracic dissection approach that enables the optical recording of descending and ascending axons within the cervical connective in tethered, behaving animals (*Chen et al., 2018*; *Hermans et al., 2021*; *Figure 1a*). Here, we combined this imaging approach with genetic tools (*Asahina et al., 2014*) that restrict the expression of neural activity reporters to the brain. This allowed us to record populations of nearly 100 DNs in individual tethered, behaving flies. During these recordings we presented olfactory stimuli and acquired behavioral data for 3D pose estimation (*Günel et al., 2019*), as well as fictive locomotor trajectories (*Moore et al., 2014*).

Using these tools, we could test the extent to which DN population activity patterns are consistent with roles in behavior modulation (*Figure 1b*), context-dependent recruitment (*Figure 1c*), and/or raw sensory signaling (*Figure 1d*). We observed that the largest fraction of DNs are active during walking. Principal component analysis (PCA) revealed diverse neural dynamics across epochs of walking. This variability reflected the activities of partially overlapping subsets of DNs that were correlated with turning and, to a far weaker extent, speed. These data support a role for DN populations in behavioral modulation. DNs are active during walking or grooming irrespective of whether the behavior was generated spontaneously or during olfactory stimulation. These data suggest a lack of strong context dependence in DN population recruitment. In the future, this finding can be further tested for its generality by examining DN population activity in the presence of other visual, mechanosensory, and olfactory cues. Notably, we did find that some DNs are specifically responsive to odors without carrying information about ongoing actions. Thus, motor circuits have access to surprisingly unfiltered sensory information. Finally, we illustrate how one can identify DNs from population recordings (*Figure 1e*). We studied a prominent pair of DNs that are asymmetrically active during antennal grooming. By using their topological and encoding properties, we could identify a sparse driver line that targets these neurons used morphological analysis (MultiColor FlpOut [MCFO] and connectomics) and propose that they are DNx01 neurons originating from the antennae (*Namiki et al., 2018*). These data provide a more expansive view of DN population activity during natural behaviors and open the door to a comprehensive mechanistic understanding of how the brain's descending signals regulate motor control.

## Results

### Recording descending neuron population activity in tethered, behaving *Drosophila*

To selectively record the activity of populations of DNs, we devised an intersectional genetic and optical approach. First, we restricted the expression of GCaMP6s (a fluorescent indicator of neural activity; *Chen et al., 2013*) and tdTomato (an anatomical fiduciary) to the supraesophageal zone of the brain (*Asahina et al., 2014*) (;*otd-nls:FLPo; R57C10-GAL4, tub>GAL80>*). We confirmed that transgene expression was restricted to cell bodies in the brain (*Figure 1f*). The axons of DNs could be seen passing through the cervical connective, targeting motor circuits within the VNC (*Figure 1g*). Thus, although our driver line lacks expression in the subesophageal zone (SEZ) (*Figure 1—figure supplement 1c*)—a brain region known to house at least 41 DNs (*Sterne et al., 2021*; *Namiki et al., 2018*) some of which can drive grooming (*Guo et al., 2022*) (DNg11, DNg12)—we could still capture the activities of a large population of DNs. Second, by performing coronal (x–z) two-photon imaging of the thoracic cervical connective (*Chen et al., 2018*), we could exclusively record DN axons (*Figure 1h*) in tethered animals while they behaved on a spherical treadmill. This imaging approach

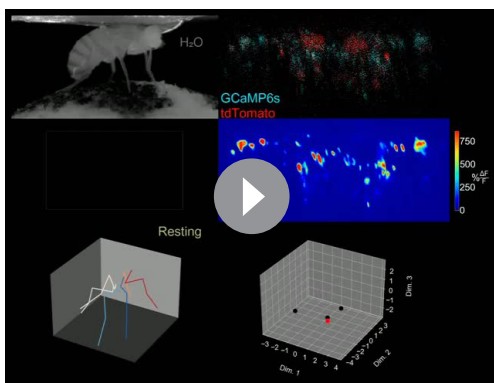

**Video 1.** Representative recording and processing of descending neuron population activity and animal behavior. (Top left) Fly behavior as seen by camera 5. Odor stimulus presentation is indicated. (Middle left) Fictive walking trajectory of the fly calculated using FicTrac. White trajectory turns gray after 2 s. (Bottom left) 3D pose of the fly calculated using DeepFly3D. Text indicates the current behavior class. (Top right) Raw two-photon microscope image after center-of-mass alignment. (Middle right) $\Delta F/F$ image after motion correction and denoising of the green channel. (Bottom right) Linear discriminant analysis-based low-dimensional representation of the neural data. Each dimension is a linear combination of neurons. The dimensions are chosen such that frames associated with different behaviors are maximally separated.

https://elifesciences.org/articles/81527/figures#video1

could also compensate for image translations during animal behavior, keeping regions-of-interest (ROIs) within the field of view (FOV). We then applied image registration (*Chen et al., 2018*)—to correct for translations and deformations—as well as image denoising (*Lecoq et al., 2021*)—to obtain higher signal-to-noise images. We confirmed that denoising does not systematically delay or prolong the temporal dynamics of neural activity (*Figure 1—figure supplement 1a and b*). Resulting images included a number of elliptical ROIs that likely represent individual large axons or possibly tightly packed groups of smaller axons (*Figure 1i*). From here on, we will interchangeably refer to ROIs as DNs or neurons. From these data, we calculated $\Delta F/F$ images (*Figure 1j*) from which manually labeled 75–95 of the most distinct and clearly visible ROIs for each animal. This resulted in high-dimensional neural activity time series (*Figure 1k*).

To test the context-dependence and sensory feedback encoding of DNs, we built an olfactometer that could sequentially present humidified air and one of two odors: ACV (an attractive odorant; *Semmelhack and Wang, 2009*) or MSC (a putatively aversive odorant; *Mohamed et al., 2019*). During experiments we alternated presentation of ACV and MSC with humid air. We performed photoionization detector (PID) measurements to confirm that our olfactometer could deliver a steady flow of air/odor with minimal mechanical perturbations during switching (*Figure 1—figure supplement 1d–f*).

Along with neural recordings and odor delivery, we quantified limb and joint positions by video recording tethered animals from six camera angles synchronously at 100 frames per second (fps) (*Figure 1l*). A seventh, front-facing camera recorded spherical treadmill rotations that were then converted into fictive locomotor trajectories using FicTrac (*Moore et al., 2014*; *Figure 1m*). Multi-view camera images (*Figure 1n*) were postprocessed using DeepFly3D (*Günel et al., 2019*) to estimate 3D joint positions (*Lobato-Rios et al., 2022*; *Figure 1o*). These data were used to train a dilated temporal convolutional neural network (DTCN) (*Whiteway et al., 2021*) that could accurately classify epochs of walking, resting, head (eye and antennal) grooming, front leg rubbing, and posterior movements (a grouping of rarely generated and difficult to distinguish hindleg and abdominal grooming movements) (*Figure 1p*, *Figure 1—figure supplement 1g*). Animals predominantly alternated between resting, walking, and head grooming with little time spent front leg rubbing or moving their posterior limbs and abdomen (*Figure 1—figure supplement 1h*). Notably, we also observed structure in our behavioral data: flies were more likely to walk after resting or generating posterior movements

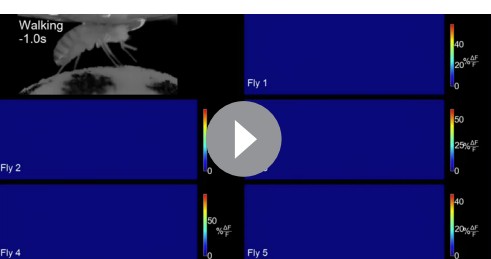

**Video 2.** Behavior-triggered average $\Delta F/F$ during walking. Averaged $\Delta F/F$ images aligned with respect to behavior onset for all walking epochs. Red circles (top left in each imaging panel) indicate the onset of behavior. When the red circle becomes cyan less than seven behavior epochs remain and the final image with more than eight epochs is shown. Shown as well is an example behavior epoch (top left) indicating the time with respect to the onset of behavior and synchronized with $\Delta F/F$ panels.

https://elifesciences.org/articles/81527/figures#video2

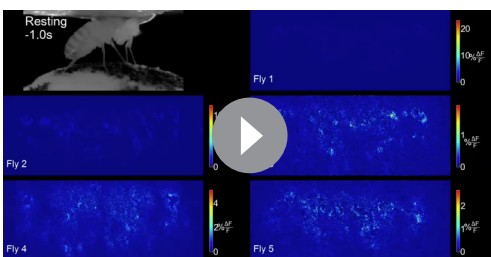

**Video 3.** Behavior-triggered average $\Delta F/F$ during resting. Averaged $\Delta F/F$ images aligned with respect to behavior onset for all resting epochs. Red circles (top left in each imaging panel) indicate the onset of behavior. When the red circle becomes cyan less than seven behavior epochs remain and the final image with more than eight epochs is shown. Shown as well is an example behavior epoch (top left) indicating the time with respect to the onset of behavior and synchronized with $\Delta F/F$ panels.
https://elifesciences.org/articles/81527/figures#video3

(*Figure 1—figure supplement 1i*). Flies also frequently performed front leg rubbing after head grooming (*Seeds et al., 2014*; *Figure 1—figure supplement 1i*). Taken together, this experimental and computational pipeline yielded a rich dataset of DN population activity and associated odor-evoked and spontaneous behaviors (*Video 1*).

## Encoding of behavior in descending neuron populations

With these data, we first asked to what extent DN populations encode—and potentially drive or regulate—each of our classified behaviors: resting, walking, head grooming, posterior movements, and front leg rubbing. The word 'encoding' has commonly been used to convey that a neuron's activity correlates with some continuously varying property of the sensory environment or an animal's motor behavior. The word 'encoding' has also previously been used in reference to DN functional properties (*Gray et al., 2010*; *Suver et al., 2016*; *Nicholas et al., 2018*; *Jaske et al., 2021*). Although fluorescence imaging has lower temporal resolution than electrophysiological recordings–making it less able to fully resolve the specificity of feature/behavioral encoding–we use the term 'encoding' for the sake of brevity.

We identified DN behavioral encoding in two ways. First, we asked how well each behavior could be predicted based on the activity of each neuron by using a linear model to quantify the extent to which a given DN's activity could explain the variance of (i.e., encode) each behavior (*Figure 2a and b*). The largest fraction (~60%) of DNs encode walking. The second largest group of DNs encode head grooming (~15%). Only a very small fraction of DNs encode resting and no neurons encode front leg rubbing or posterior movements (*Figure 2c*). However, some of these behaviors were very infrequent (posterior movements) or of short duration (front leg rubbing) (*Figure 1—figure supplement 1h*), weakening the power of our analysis in these cases. As well, although none of the DNs *best* explained posterior movements and front leg rubbing out of all behaviors we observed that DNs encoding walking also encoded posterior movements. Similarly, DNs encoding head grooming also encoded front leg rubbing (*Figure 2b*). This may be due to the strong sequential occurrence of these pairs of behaviors (*Figure 1—figure supplement 1i*) and the long decay time constant of GCaMP6s (~1 s; *Chen et al., 2013*), resulting in elevated calcium signals for the subsequent behavior in the pair. To resolve the extent to which these DNs truly encode one behavior versus the other we performed a more narrow linear regression analysis. We used equal amounts of data from this pair of sequential behaviors and calculated the neural variance (rather than the behavioral variance) uniquely explained by these two behaviors alone. These analyses confirmed that DNs predominantly encode walking rather than posterior movements (*Figure 2—figure supplement 1a*) and head grooming rather than front leg rubbing (*Figure 2—figure supplement 1b*).

We next asked to what extent DNs encode—and presumably drive—the kinematics of joints,

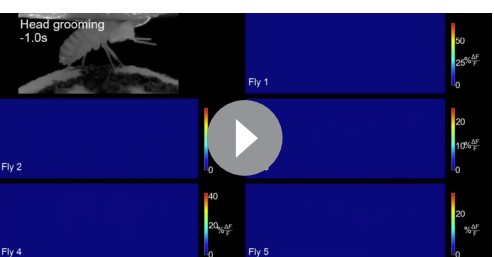

**Video 4.** Behavior-triggered average $\Delta F/F$ during head grooming. Averaged $\Delta F/F$ images aligned with respect to behavior onset for all head grooming epochs. Red circles (top left in each imaging panel) indicate the onset of behavior. When the red circle becomes cyan less than seven behavior epochs remain and the final image with more than eight epochs is shown. Shown as well is an example behavior epoch (top left) indicating the time with respect to the onset of behavior and synchronized with $\Delta F/F$ panels.
https://elifesciences.org/articles/81527/figures#video4

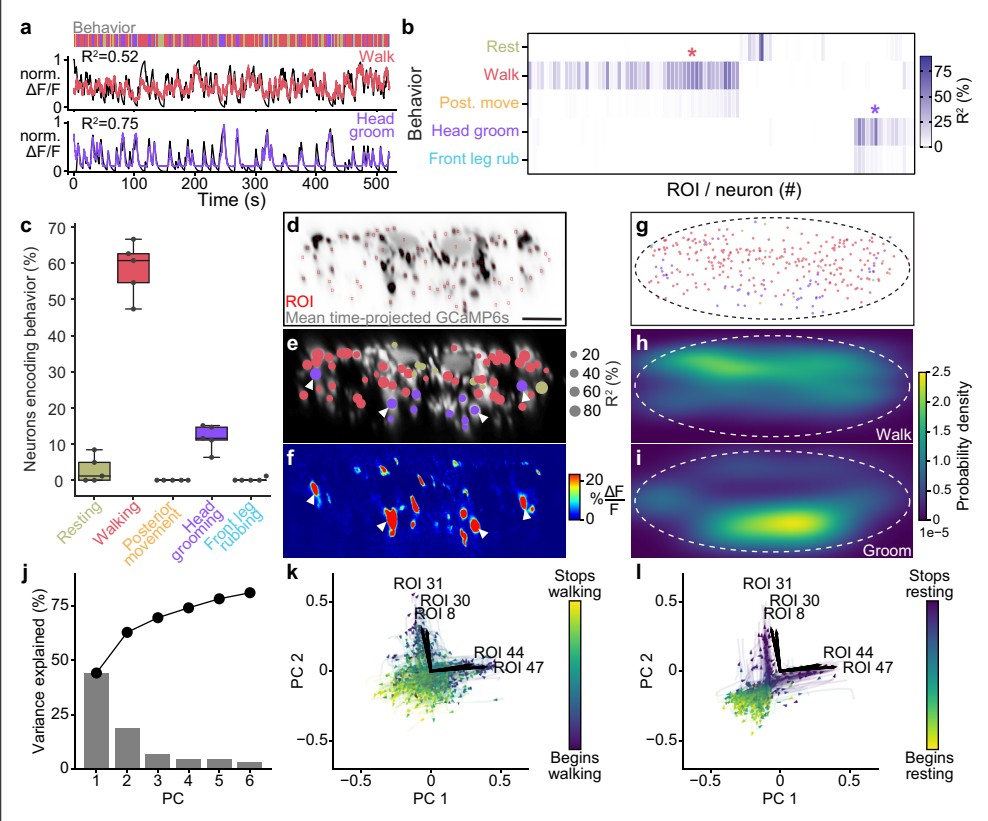

**Figure 2.** Encoding of behavior in descending neuron (DN) populations. (**a**) Shown for walking (top) and head grooming (bottom) are the activity (normalized and cross-validation predicted $\Delta F/F$) of individual walk- and head groom-encoding DNs (red and purple lines), as well as predicted $\Delta F/F$ traces derived by convolving binary behavior regressors with a calcium response function (crf) (black lines). The output of the behavior classifier is shown (color bar). (**b**) The cross-validation mean of behavioral variance explained by each of 95 DNs from one animal. Colored asterisks are above the two DNs illustrated in panel (**a**). (**c**) The percentage of DNs encoding each classified behavior across five animals. Box plots indicate the median, lower, and upper quartiles. Whiskers signify furthest data points. (**d**) Mean time projection of GCaMP6s fluorescence over one 9 min recording. Image is inverted for clarity (high mean fluorescence is black). Manually identified DN regions of interest (ROIs) are shown (red rectangles). Scale bar is 10 μm. Panels (**d–i**) share the same scale. (**e**) DNs color-coded (as in panel **c**) by the behavior their activities best explain. Radius scales with the amount of variance explained. Prominent head groom-encoding neurons that are easily identified across animals are indicated (white arrowheads). (**f**) Behavior-triggered average $\Delta F/F$ image for head grooming. Prominent head grooming DNs identified through linear regression in panel (**e**) are indicated (white arrowheads). (**g**) Locations of DNs color-coded by the behavior they encode best. Data are from five animals. (**h, i**) Kernel density estimate based on the locations of (**h**) walking or (**i**) head grooming DNs in panel (**g**). (**j**) Amount of variance explained by the principal components (PCs) of neural activity derivatives during walking. (**k, l**) Neural activity data during (**k**) walking and (**l**) resting evolve on two lobes. The PC embedding was trained on data taken during walking only. Colored lines indicate individual epochs of (**k**) walking and (**l**) resting. Time is color-coded and the temporal progressions of each epoch is indicated (arrowheads). Note that color scales are inverted to match the color at transitions between walking and resting. Black arrows indicate ROIs with high PC loadings. ROI number corresponds to the matrix position in panel (**b**). For their locations within this fly's cervical connective, see *Figure 2—figure supplement 4d*.

The online version of this article includes the following figure supplement(s) for figure 2:

**Figure supplement 1.** Disentangling the relative encoding of frequently sequential behavior pairs.

**Figure supplement 2.** Encoding of behavior in descending neuron (DN) populations across individual animals.

**Figure supplement 3.** Neural variance explained by distinct kinematic features.

**Figure supplement 4.** Principal component (PC) analysis of neural activity during walking and resting across individual animals.

limbs, or limb pairs rather than behaviors. To test this possibility, we quantified how much better neural activity could be predicted from joint angles rather than from behavior. Specifically, we computed the amount of variance in DN activity that could be uniquely explained by subgroups of joint angles but not behavior categories or any of the remaining joint angles. Separating joint angles into groups (all, pairs, or individual legs) allowed us to probe the possibility that some neurons might control pairs of legs or individual legs and helped to mitigate the effect of correlations between joint angles within individual legs. Because of the rapid movements of each leg ($5-20\,\mathrm{Hz}$; *Ravbar et al., 2021*; *Mendes et al., 2013*) and the long decay time of our calcium indicator, we also convolved joint angle and behavior regressors with a calcium response function (crf) kernel. We found that joint angles can only very marginally improve the prediction of neural activity beyond simply using behavior regressors (*Figure 2—figure supplement 3*). Thus, DN populations largely encode high-level behaviors suggesting that they delegate low-level kinematic control to downstream circuits in the VNC.

## The spatial organization of descending neuron encoding

Previous morphological analyses demonstrated a clear organization of *Drosophila* DN projections within the VNC (*Namiki et al., 2018*). This is likely linked to the distinct functional partners of DNs that regulate limb-dependent (e.g., walking and grooming) versus wing-dependent (e.g., flight and courtship display) behaviors. To further explore the relationship between function and topology, we next asked to what extent we might observe a relationship between a DN's encoding of behavior and its axon's position within the cervical connective (*Figure 2d*). We found that DNs encoding walking are spread throughout the dorsal connective (*Figure 2e, g and h*). On the other hand, head groom-encoding DNs are predominantly in the ventral connective (*Figure 2g and i*) including two prominent pairs—lateral and medial-ventral (*Figure 2e*, white arrowheads)—whose activities explain the largest amount of variance in head grooming across animals (*Figure 2—figure supplement 2c*). Surprisingly, we also observed DNs that encode resting. These were located medially, close to the giant fibers, as well as in the lateral extremities of the connective (*Figure 2e*, olive circles). We speculate that rest-encoding DNs may suppress other behaviors or could actively drive the tonic muscle tone required to maintain a natural posture.

We next performed a complementary analysis to further examine the functional–topological organization of DNs in the connective. Specifically, we generated behavior-triggered averages of $\Delta F/F$ images. The results of this approach were only interpretable for frequently occurring behaviors so here we focused on walking, resting, and head grooming (*Figure 1—figure supplement 1h*; *Videos 2–4*). Across animals, we consistently observed DNs in the dorsal connective encoding walking, and two pairs of ventral DNs encoding head grooming (*Figure 2f*, *Figure 2—figure supplement 2d*). By contrast, rest-encoding DNs were located in less consistent locations within the connective. These findings confirm that DN populations for walking and head grooming are largely spatially segregated, a feature that may facilitate the identification of specific cells from DN population recordings across animals.

## Descending neuron population dynamics suggest more nuanced feature encoding

Thus far we have observed that DN subpopulations encode distinct behaviors and that, by far, the largest fraction encode walking. However, locomotion is not monolithic. It continuously varies in speed and direction within a single walking trajectory. Thus, the large number of DNs active during walking may represent an aggregate of subpopulations that are differentially engaged during distinct locomotor modes. To address this hypothesis, we first closely examined the temporal structure of DN population activity dynamics only during walking epochs. We asked to what extent there is variability and structure in population activity that could potentially support the modulatory encoding of walking speed, forward/backward, or turning.

As in a similar analysis of *Caenorhabditis elegans* population dynamics (*Kato et al., 2015*), we calculated the temporal derivative of each DN's activity and then performed principal component analysis (PCA) on these time series. We found that the first two PCs can explain upwards of 60% of the variance in DN population activity during walking (*Figure 2j*, *Figure 2—figure supplement 4a*). Visualizing 2D trajectories (PC 1 and PC 2 subspace) of DN activity during individual walking bouts revealed that they move primarily along two directions (*Figure 2k*, *Figure 2—figure supplement*

*4b*). These directions are even more clear for resting data (embedded within the same PC space) just before the fly began to walk (*Figure 2l*, *Figure 2—figure supplement 4c*).

To identify individual DNs that most heavily influence this dynamical divergence we next found those with the largest PC loadings. These neurons' activities most strongly influence the position of population activity in PC space (*Figure 2k and l*, *Figure 2—figure supplement 4b and c*). Consistently, also in flies with a less clear divergence in neural trajectories, we found subsets of DNs whose activities correspond to one of these two directions. By examining the positions of their axons we observed that they are spatially segregated on opposite sides of the connective (*Figure 2—figure supplement 4d*, cyan arrowheads).

## Descending neurons that encode walking include spatially segregated turn-encoding clusters

The divergence of population dynamics and spatial segregation of associated neurons led us to hypothesize that subsets of walk-encoding DNs might preferentially become active during left and right turning. Alternatively, they might encode fast versus slow walking speeds (*Bidaye et al., 2020*). Studies in other insects and vertebrates have shown that DNs can play a modulatory role by regulating turning and speed during locomotion (*Capelli et al., 2017*; *Heinrich, 2002*). As well, in *Drosophila*, the activation of DNp09 neurons can drive forward walking (*Bidaye et al., 2020*). Recordings from sparse sets of DNa01 (*Chen et al., 2018*; *Rayshubskiy, 2020*) and DNa02 (*Rayshubskiy, 2020*) neurons also show turn encoding (i.e., steering).

Therefore, we next tested whether variability in the activity of DNs might reflect fine-grained encoding of turning and speed during forward walking. We did not analyze backward walking due to its scarcity (*Figure 3—figure supplement 1a*), brevity (*Figure 3—figure supplement 1b*), and minimal dynamic range (*Figure 3—figure supplement 1c*) in our experimental data. Specifically, we quantified the degree to which DN population activity could uniquely explain yaw (turning) or pitch (speed) angular velocity of the spherical treadmill. Both of these time-series data were convolved with a crf to account for slow calcium indicator decay dynamics. To capture information about turning and walking speed that could not simply be explained by whether the fly was walking or not we compared the explained variance of our neuron-based ridge regression to a model predicting these features from just a binary walking regressor and shuffled neural data (*Figure 3a*). Neural activity could explain a great deal of variance in turning and, to a lesser extent, speed. Here, the absence of speed encoding may be because the binary walking regressor alone can partially predict speed variance from transitions between resting and walking ($R^2 = 25\%$). Therefore, we refined our analysis by only using data from walking epochs and calculating the amount of turning and speed variance that could be explained using neural activity. In this manner, we confirmed that neural activity can uniquely explain turning to a greater extent (~60%) than it can explain walking speed (~30%) (*Figure 3b*).

We next investigated which individual or groups of neurons contribute to the prediction of turning and walking speed. To do this we only used the activity of one neuron at a time in our model (by contrast, above we used all neurons). Among DNs that encode walking, we found specific DNs that strongly explain right or left turning. By contrast, a more distributed set of DNs weakly encode walking speed (*Figure 3c*, *Figure 3—figure supplement 2a*). Having identified clusters of DNs encoding right (red) and left (blue) turning, we next investigated whether there might be groups encoding other walking features. Among neurons that best explain walking we again observed clusters for turning but no prominent clusters for speed (*Figure 3d*, *Figure 3—figure supplement 2b*). This was also reflected in the amount of variance explained: across animals some walk-encoding DNs also strongly encoded turning (*Figure 3e*) whereas these DNs only encoded a tiny fraction of the variance in walking speed (*Figure 3f*).

Simple models for locomotor control (*Braitenberg, 1986*) suggest that turning can be controlled by the relative activities of DNs on one side of the brain versus the other. This is supported by studies showing that flies generate turning during asymmetric activation or asymmetric activity of DNa01 neurons (*Chen et al., 2018*; *Rayshubskiy, 2020*) and MDNs (*Sen et al., 2017*). To examine the degree to which this spatial asymmetry extends beyond pairs of neurons to much larger DN populations we quantified the spatial location of turn-encoding DNs in the cervical connective. We found both ipsi- and contralateral turn-encoding DNs on both sides of the connective (*Figure 3g*, *Figure 3—figure supplement 2c*) but a clear ipsilateral enrichment (*Figure 3h*, *Figure 3—figure supplement 2d*). Many

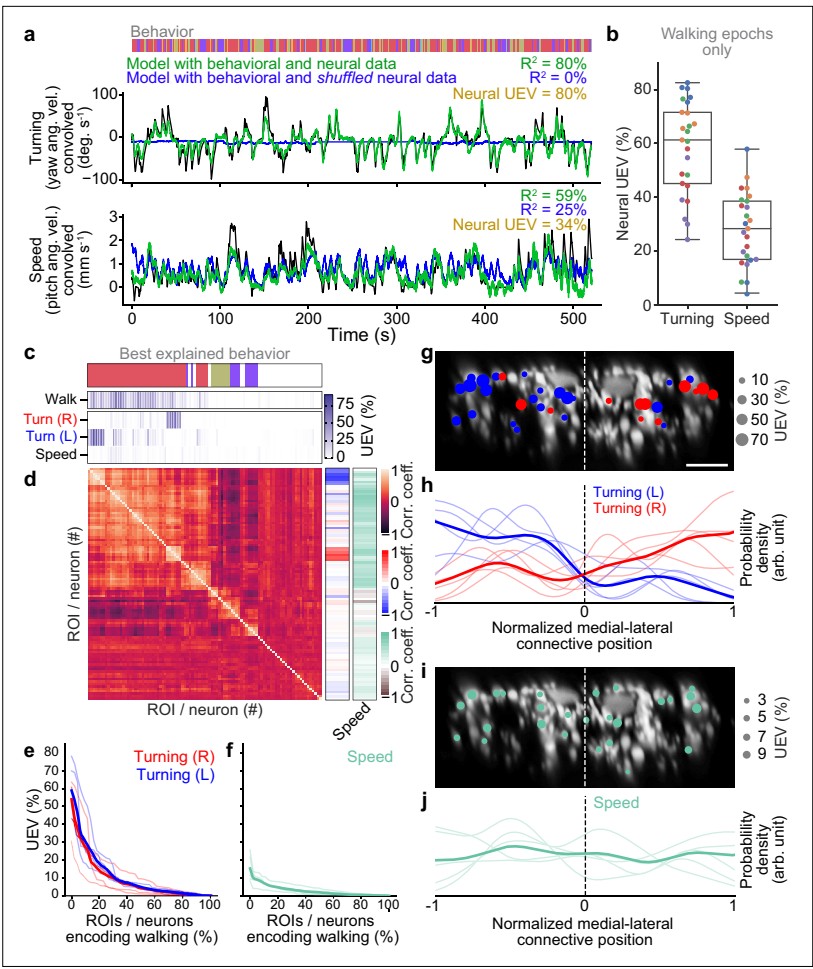

**Figure 3.** Turning and speed encoding in descending neuron (DN) populations. (**a**) Predictions of (top) turning and (bottom) walking speed modeled using convolved behavior regressors and all neurons in one animal. Shown are predictions (green) with all regressors intact or (blue) with neural data shuffled across time. Indicated are $R^2$ values obtained by comparing predicted and real (black) turning and walking speed. These are subtracted to obtain neural unique explained variance (UEV). The fly's behavior throughout the recording is indicated (color bar). (**b**) UEV obtained only using data taken during walking, thus accounting for trivial explanations of speed and turning variance resulting from transitions between resting and walking. Shown are data from five trials each for five flies (color-coded). (**c**) UEV of each DN from one animal for walking speed or left and right turning ordered by clustering of Pearson's correlation coefficients in panel (**d**). Walking $R^2$ values are the same as in **Figure 2b** but reordered according to clustering. The models for turning and walking speed were obtained using behavior regressors as well as neural activity. To compute the UEV, activity for a given neuron was shuffled temporally. The behavior whose variance is best explained by a given neuron is indicated (color bar). (**d**) Pearson's correlation coefficient matrix comparing neural activity across DNs ordered by clustering. Shown as well are the correlation of each DN's activity with right, left, and forward walking (right). (**e, f**) UEV for (**e**) turning or (**f**) speed for DNs that best encode walking. Neurons are sorted by UEV. Shown are the distributions for individuals (translucent lines), and the mean across all animals (opaque line). (**g**) Locations of turn-encoding DNs (UEV > 5%), color-coded by preferred direction (left, blue; right, red). Circle radii scale with UEV. Dashed white line indicates the approximate midline of the cervical connective. Scale bar is 10 μm for panels (**g**) and (**i**). (**h**) Kernel density estimate of the distribution of turn encoding DNs. Shown are the distributions for individuals (translucent lines), and the mean distribution across all animals (opaque lines). Probability densities are normalized by the number of DNs along the connective's medial–lateral axis. (**i**) Locations of speed encoding DNs (UEV > 2%). Circle radii scale with UEV. Dashed white line indicates the approximate midline of the cervical connective. (**j**) Kernel density estimate of the distribution of speed encoding DNs. Shown are the distributions for individuals (translucent lines) and the mean distributions across all animals (opaque lines). Probability densities are normalized by the number of DNs along the connective's medial–lateral axis.

The online version of this article includes the following figure supplement(s) for figure 3:

*Figure 3 continued on next page*

*Figure 3 continued*

**Figure supplement 1.** Backward walking is infrequent and brief.

**Figure supplement 2.** Turning and speed encoding in descending neuron (DN) populations across individual animals.

of the DNs encoding turning had high PC loading (*Figure 2—figure supplement 4d*) revealing that turning contributes heavily to the variance in DN population dynamics during walking. By contrast, DNs encoding walking speed were more homogeneously distributed across the connective with no clear spatial enrichment (*Figure 3i and j*, *Figure 3—figure supplement 2e and f*). Overall, these data support the notion that, during walking, DN population activity largely varies due to turn-related modulation rather than shifts in walking speed.

## Descending neurons are active during behaviors irrespective of olfactory context

Beyond a modulatory role, the large number of DNs active during walking could reflect context dependence: specific subpopulations may only be engaged as a function of sensory context. The possibility of recruiting separate pools of DNs for walking is supported by the observation that an attractive odor, ACV, decreases resting and increases walking (*Figure 4—figure supplement 1a*), increases forward walking speed (*Figure 4—figure supplement 1b*), and reduces turning (*Figure 4—figure supplement 1c and d*).

To address the extent to which subgroups of DNs are recruited depending on olfactory context, we studied the amount of walking or head grooming variance explained by each DN using only data acquired during exposure to either humidified air, ACV, or MSC—rather than analyzing all walking epochs as in our previous analysis. Humidified air data were subsampled to match the smaller amount of data available for ACV and MSC presentation. We found that largely the same DNs were highly predictive of walking (*Figure 4—figure supplement 2a*) and head grooming (*Figure 4—figure supplement 2b*) irrespective of olfactory context. Only a very small fraction of DNs were differentially recruited during odor presentation (*Figure 4—figure supplement 2a and b*, black asterisks, two-sided Mann–Whitney *U*-test on cross-validation folds). Of these four DNs with different recruitment, three achieve significance because they have only a few values distinct from zero in a single trial. The overall explained variance is also very small (*Figure 4—figure supplement 2c*). These data suggest that changing odor context alters action selection and locomotor kinematics but does not shift the identity of active DN subpopulations driving behavior.

## Descending neurons exhibit raw odor encoding

Although walk- and head groom-encoding DN populations are recruited irrespective of odor context, it has been shown that a fictive odor (i.e., optogenetic activation of *Orco>CsChrimson*) can activate DNa02 neurons in immobile animals (*Rayshubskiy, 2020*). This implies that DNs may encode the presence and identity of real odors. To examine the extent of this raw sensory encoding we trained and cross-validated linear discriminant odor classifiers using neural residuals (i.e., the neural activity remaining after subtracting activity that could be predicted using a model based on crf convolved behavior regressors). These residuals allowed us to control for the fact that odor presentation also modulates behavioral statistics (*Figure 4—figure supplement 1a–c*).

Classification using neural residuals performed significantly better than classification using behavioral information alone ($p < 10^{-5}$ for a two-sided Mann–Whitney *U*-test; *Figure 4a*). This reveals raw odor encoding within DN populations. However, this might result from many neurons with weak, complementary odor encoding or a few neurons with strong odor encoding. To distinguish between these possibilities, we next identified which DNs encode olfactory signals. We predicted each neuron's activity using regressors for behavior and the presence of each odor. The more intuitive approach of modeling neural activity by just using odor regressors does not account for the confound that behaviors are also modulated by specific odors. Therefore, we computed the amount of neural variance that could uniquely be explained by the presence of an odor and none of the behavior variables. We found that the activity of a few DNs could be uniquely explained by each of the odors (*Figure 4b*, asterisks). In these neurons, clear activity peaks coincided with the presence of MSC (*Figure 4c*) or ACV

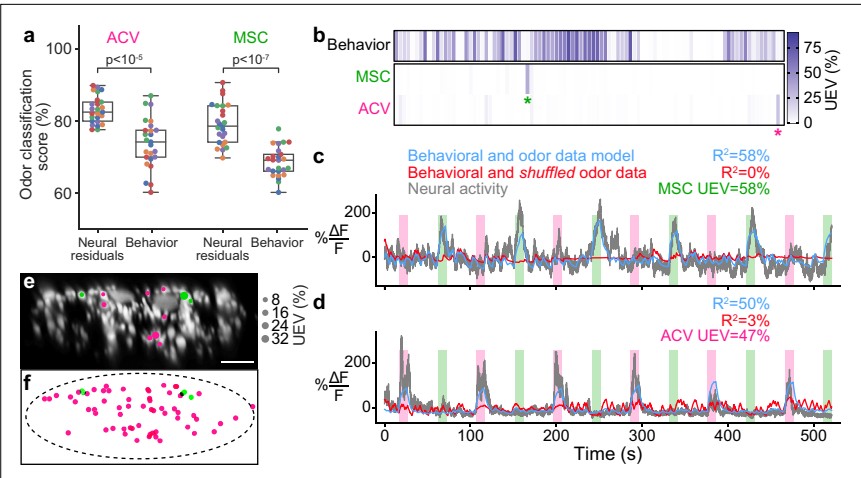

**Figure 4.** Odor encoding in descending neuron (DN) populations. (**a**) Neural residuals—obtained by subtracting convolved behavior regressors from raw neural data—can predict the presence of an odor significantly better than behavior regressors convolved with a calcium response function ('Behavior'). Two-sided Mann–Whitney $U$-test. The classification score was obtained using a linear discriminant classifier with cross-validation. Shown are five trials for five animals (color-coded). (**b**) Matrix showing the cross-validated ridge regression unique explained variance (UEV) of a model that contains behavior and odor regressors for one animal. The first row ('Behavior') shows the composite $R^2$ for all behavior regressors with odor regressors shuffled. The second and third rows show the UEVs for regressors of the odors methyl salicylate (MSC) or apple cider vinegar (ACV), respectively. Colored asterisks indicate neurons illustrated in panel (**a**). (**c, d**) Example DNs best encoding (**c**) MSC or (**d**) ACV, respectively. Overlaid are traces of neural activity (gray), row one in the matrix (blue), and row one with odor data shuffled (red). (**e, f**) Locations of odor encoding neurons in (**e**) one individual and (**f**) across all five animals. Scale bar is 10 μm.

The online version of this article includes the following figure supplement(s) for figure 4:

**Figure supplement 1.** Odor-modulated behaviors and encoding in descending neuron (DN) populations across individuals.

**Figure supplement 2.** Largely identical descending neuron (DN) populations are recruited during walking and head grooming irrespective of odor context.

(*Figure 4d*). Notably, there appears to be no overlap between the DNs encoding MSC or ACV within individual animals (*Figure 4—figure supplement 1e*). MSC encoding neurons were found dorsally on the lateral sides of the giant fibers in the connective while ACV encoding neurons were more broadly dispersed (*Figure 4e and f*).

## Identifying individual descending neurons from population recordings

Until now we have demonstrated that DN populations exhibit heterogeneous encoding: large, distributed groups encode walking and, by contrast, a few prominent pairs encode head grooming. Determining how these subpopulations control adaptive behavior is an important future challenge that will require a comprehensive approach examining phenomena ranging from global DN population dynamics down to the synaptic connectivity of individual DNs. The recent generation of hundreds of sparse transgenic driver lines (*Jenett et al., 2012*; *Namiki et al., 2018*) and several connectomics datasets (*Phelps et al., 2021*) suggests that this bridging of mechanistic scales may soon be within reach in *Drosophila*.

To illustrate how this might be accomplished, we aimed to identify specific DNs within our population imaging dataset. Specifically, while analyzing head grooming DNs we noticed a large pair of ventral neurons (*Figure 5a*) that sometimes exhibited asymmetric activity (*Figure 5b*, gray arrowheads) when flies appeared to touch one rather than both antennae (*Video 5*). To quantify this observation, we replayed limb 3D kinematics in NeuroMechFly, a biomechanical simulation of *Drosophila* (*Lobato-Rios et al., 2022*; *Figure 5c*). By detecting leg-antennal collisions as a proxy for antenna deflection, we found that occasional asymmetries (*Figure 5d*) did coincide with asymmetric activity in corresponding neural data (*Figure 5b*, purple traces). These results suggested that this pair of DNs encodes mechanosensory signals associated with antennal deflections.

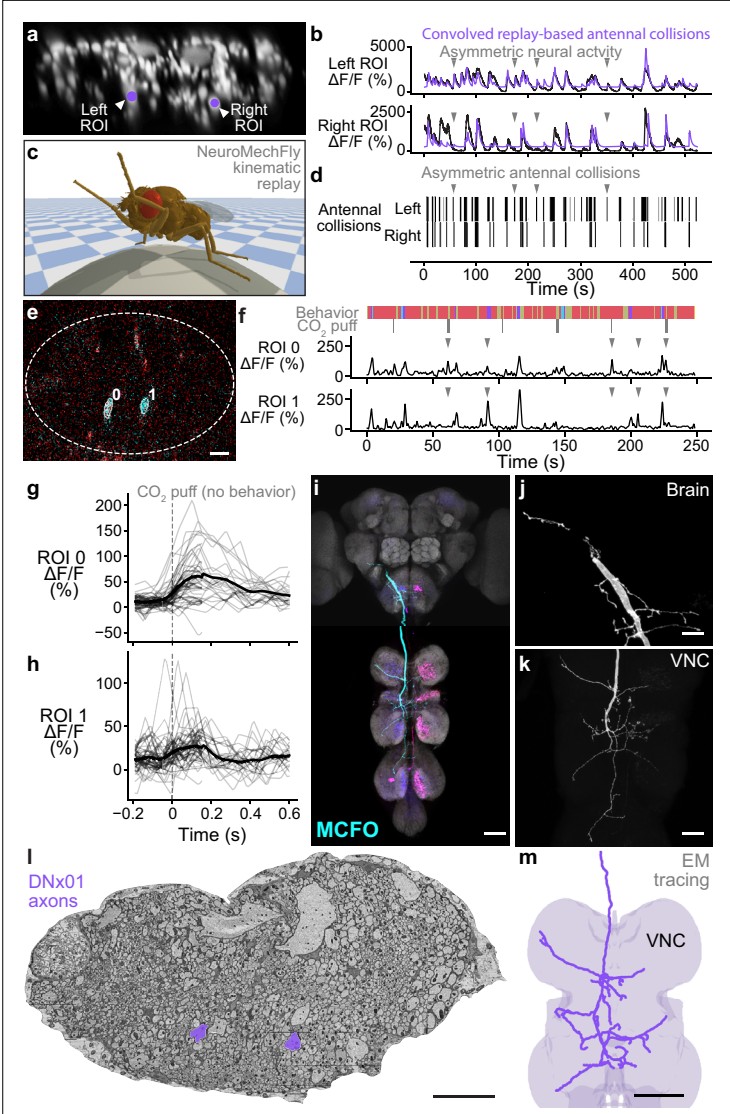

**Figure 5.** Identifying a pair of antennal deflection-encoding descending neurons (DNs) from population recordings. (**a**) A pair of head groom-encoding DNs (purple circles and white arrowheads) can be identified from DN population recordings based on their shapes, locations, and activity patterns. (**b**) Example $\Delta F/F$ traces (black) of DNs highlighted in panel (**a**). Sample time points with bilaterally asymmetric neural activity are indicated (gray arrowheads). Overlaid is a prediction of neural activity derived by convolving left and right antennal collisions measured through kinematic replay in the NeuroMechFly physics simulation (purple). (**c**) Kinematic replay of recorded joint angles in NeuroMechFly allow one to infer antennal collisions from real, recorded head grooming. (**d**) Left and right antennal collisions during simulated replay of head grooming shown in panel (**b**). Sample time points with bilaterally asymmetric collisions are indicated (gray arrowheads). (**e**) Two-photon image of the cervical connective in a R65D11>OpGCaMP6f, tdTomato animal. Overlaid are regions of interest (ROIs) identified using AxoID. The pair of axonal ROIs are in a similar ventral location and have a similarly large relative size like those seen in DN population recordings. Scale bar is 5 µm. (**f**) Sample neural activity traces from ROIs 0 and 1. Bilaterally asymmetric neural activity events (gray arrowheads), behaviors (color bar), and $CO_2$ puffs directed at the antennae (gray bars) are indicated. (**g, h**) $CO_2$ puff-triggered average of neural activity for ROIs (**g**) 0 and (**h**) 1. Only events in which animals did not respond with head grooming or front leg rubbing were used. Stimuli were presented at $t = 0$. Shown are individual responses (gray lines) and their means (black lines). (**i**) Confocal volume z-projection of MultiColor FlpOut (MCFO) expression in an R65D11-GAL4 animal. Cyan neuron morphology closely resembles DNx01 (**Namiki et al., 2018**). Scale bar is 50 µm. (**j, k**) Higher-magnification MCFO image, isolating the putative DNx01 from panel (**i**), of the (**j**) brain and (**k**) ventral nerve cord (VNC). Scale bars are 20 µm. (**l**) The locations of axons in the cervical connective (purple) from neurons identified as DNx01. Scale bar is 10 µm. (**m**) Manual reconstruction of a DNx01 from panel (**l**). Scale bar is 50 µm.

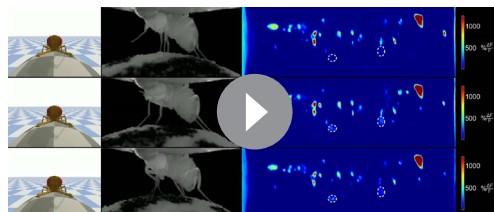

**Video 5.** Asymmetric activity in a pair of DNx01s is associated with asymmetric leg-antennal collisions during antennal grooming. Three head grooming epochs in the same animal having leg contact with (top) primarily the left antenna, (middle) both antennae, or (bottom) primarily the right antenna. Shown are corresponding (right) neural activity $\Delta F/F$ images (note putative DNx01s in dashed white circles), (middle) behavior videos (camera 2), and (left) leg-antennal collisions (green) during kinematic replay of 3D poses in the NeuroMechFly physics simulation. Video playback is 0.25× real time.

https://elifesciences.org/articles/81527/figures#video5

To further reveal the identity of these DNs, we examined data from our functional screen of sparse Gal4 and split Gal4 driver lines (*Chen, 2022*). In this dataset, we observed similar asymmetric activity during antennal grooming in R65D11-GAL4. Coronal (x–z) two-photon imaging in R65D11 animals expressing OpGCaMP6f and tdTomato shows axons that are similarly large and ventromedially located within the cervical connective (*Figure 5e*). These also produce asymmetric activity during antennal grooming (*Figure 5f*). This suggests that these neurons may report something unique to head grooming (e.g., coincident front limb movements) or simply antennal deflection. To distinguish between these possibilities we analyzed neural responses to $CO_2$ puff stimulation of the antennae while discarding data with resulting head grooming or front leg rubbing to ensure that the antennae were not touched by the legs. We measured an increase in the activity of both DNs upon puff stimulation (*Figure 5g and h*) suggesting that, like the neurons recorded in DN populations, R65D11 neurons also encode sensory signals—antennal deflection—rather than behavior.

To confirm that these sparse neurons are DNs, we next performed MCFO (*Nern et al., 2015*) and confocal imaging of their morphologies. R65D11 drives expression in several neurons. However, we found similarly large axonal projections from only one set of neurons that descend from the brain to the VNC (*Figure 5i*, cyan). Close examination of these neurites in the brain (*Figure 5j*) and VNC (*Figure 5k*) revealed a striking resemblance to the reported structure of DNx01 neurons (*Namiki et al., 2018*) with cell bodies outside of the brain—putatively in the antennae and enabling antennal mechanosensing.

These results enable the analysis of synaptic connectivity in identified DNs. To illustrate this, based on their unique location and size, we identified DNx01s in a VNC electron microscopy dataset (*Phelps et al., 2021*; *Figure 5l*) via manual reconstruction and observed a striking morphological similarity to R65D11 DNs (*Figure 5m*). From this reconstruction, once the full VNC connectome becomes available, one may identify synaptic partners of DNx01s to further understand how they contribute to controlling antennal grooming and other behaviors. Taken together, these data suggest a possible road map for using functional, topological, and morphological data to decipher the cellular identity of individual DNs from population recordings.

## Discussion

Here, by combining genetic and optical imaging approaches, we recorded the behavioral and sensory encoding of DN populations in behaving *Drosophila*. Although electrophysiology provides higher temporal resolution (e.g., being capable of reporting spike timing; *von Reyn et al., 2014*), neural population imaging serves an important complementary role in capturing the proportion and spatial locations of co-active neurons. Electrophysiological recordings and calcium imaging of sparse DN driver lines can more easily enable links to be made between neural encoding and cellular identity. However, these approaches suffer from two major disadvantages. First, there exist split-Gal4 driver lines for only a small fraction of DNs (*Namiki et al., 2018*). Second, determining the encoding of an equivalent number of neurons requires many more sparse neural recordings than population imaging experiments. Therefore, by focusing on population imaging our study allowed us to more rapidly survey the encoding of a large number of DNs. Using this approach we found that most recorded DNs encode walking. A smaller number are active during head grooming and resting. We did not find DNs encoding posterior movements possibly due to the infrequency of this behavior. We also did not identify neurons that are active during multiple behaviors. This suggests that each ROI consists of

individual neurons or that, if an ROI contains many neurons, they all show similar encoding. Subsets of walk-encoding DNs were also strongly active during turning: they were at higher density on the ipsilateral half of the cervical connective with respect to turn direction. By contrast, DNs distributed throughout the connective very weakly encoded walking speed. However, we caution that the small range of walking speeds in our data—flies accelerate rapidly from resting to walking and vice versa— may mask stronger speed encoding. The partial overlap between turn- and speed-encoding DNs leaves open the possibility that neurons simultaneously encode these two properties in a differential steering fashion. Notably, we did not observe any DNs that are only active during transitions between multiple behaviors. However, the fly makes fast transitions suggesting that the signal-to-noise and temporal resolution of our approach may not be sufficient to identify such neurons—higher temporal resolution electrophysiological recordings would be required to confirm the absence of DN encoding for behavioral transitions or for precise limb kinematics. Our dataset consists of natural walking and grooming behaviors. Therefore, some joint angles are more frequent than others (*Figure 1—figure supplement 2*). In future work, along with performing higher temporal resolution recordings, a larger range of possible joint angles could be explored by eliciting rarer and more complex behaviors including reaching or tethered locomotion over rugged terrain.

The encoding—and presumptive control—of walking by large numbers of DNs supports the notion that the brain tightly regulates locomotion. In contrast to walking, head grooming is encoded by far fewer neurons in our dataset. This may be because grooming limb movements are more stereotyped and thus may rely more heavily on controllers within the VNC (a notion that is supported by the ability of headless flies to perform spontaneous grooming; *Harris et al., 2015*). Other studies have also shown brain-wide activity during walking but not during other behaviors (*Aimon et al., 2019*; *Schaffer et al., 2021*; *Brezovec et al., 2022*). This difference may arise because adaptive locomotion—to avoid obstacles (*Tanaka and Clark, 2022*), cross gaps (*Triphan et al., 2010*), and court potential mates (*Coen et al., 2014*)—depends heavily on the brain's descending signals. Thus, we hypothesize that, although a core set of command neurons can drive both walking and grooming, more DNs are additionally engaged during walking to allow for more flexible navigation in continuously changing and complex environments. Interestingly, many ascending neurons (ANs) have also been shown to encode walking (*Chen, 2022*; *Fujiwara et al., 2022*) with a large fraction projecting to the gnathal ganglia (GNG), a brain region that is also heavily innervated by DNs (*Namiki et al., 2018*). Thus, we speculate that ANs and DNs may be directly connected—possibly to mediate action selection (*Mann et al., 2013*; *Bidaye et al., 2014*)—and that this may lead to similar functional encoding.

Because of the large number of DNs involved in walking we hypothesized that subsets might represent parallel channels that are recruited depending on sensory context. For example, there may be DN subpopulations that drive walking in the presence of attractive odors and others engaged in the presence of aversive odors. This notion is supported by studies showing that optogenetic activation of a large variety of DNs elicits only a small set of stereotyped behaviors (*Cande, 2018*). However, our population imaging data do not support this notion: largely the same DNs encode walking and head grooming irrespective of olfactory context.

A nonbehavioral role for DNs has also been suggested by previous studies showing that the perception of a fictive odor modulates the activity of specific DNs involved in steering in immobile flies (*Rayshubskiy, 2020*). In line with this, we also identified DNs encoding odors and not behavior. Notably, the two odors we presented—ACV and MSC—were encoded by distinct DNs. Extrapolating beyond ACV and MSC, it seems unlikely that DNs encode many individual odors with high specificity: the number of DNs is far smaller than what would be required to cover an enormous olfactory space. Instead, we speculate that DN odor-encoding may represent classes like attractive versus aversive odors or the valence of sensory inputs in general. Where odor information is conveyed to within downstream motor circuits and for what purpose is a fascinating subject for future study.

Many fewer neurons encode head grooming as opposed to walking. Because our transgenic strain does not drive expression in SEZ neurons we could not record several DNs whose activation has been shown to be sufficient to drive antennal grooming (aDN), front leg rubbing (DNg11), or both head grooming and front leg rubbing (DNg12) (*Guo et al., 2022*). Thus, we expect that with the addition

of these SEZ DNs the apparent dichotomy that many neurons encode walking and fewer encode grooming may become less pronounced. Nevertheless, groom-encoding DNs were notable in that they could be more reliably identified across individual animals. Among our head groom-encoding DNs a pair passing through the ventral cervical connective appear to encode limb contact during antennal grooming as well as puff-dependent antennal deflections. We speculate that mechanosensory signals from the antennae may be used for feedback control in the VNC: being aware of whether the antennae are touched while grooming may allow for a continuous modulation of grooming kinematics and force application by the front legs. This may be a conserved control mechanism as similar DNs have been identified in the blow fly (*Nässel et al., 1984*).

Our morphological and physiological evidence suggests that these antennal mechanosensory signals are provided by DNx01s, a subset of bilateral campaniform sensillum (bCS) neurons that are also found on the legs and form major presynaptic connections to fast motor neurons (*Phelps et al., 2021*). Thus, DNx01s may have a role beyond feedback control during grooming. This is also implied by the large size of DNx01 axons and their high sensitivity to puff-mediated antennal deflection. Because other DNs with large axons (e.g., giant fiber neurons) are often implicated in fast reflexive movements (*King and Wyman, 1980*; *Azevedo et al., 2020*), DNx01s may be used to respond to, for example, strong gusts of wind that initiate a stance stabilization reflex. Testing this hypothesis will require the creation of split-Gal4 driver lines that precisely and uniquely target DNx01 neurons.

Our analysis of DNx01 illustrates a potential road map for combining functional, topological, and morphological data to uncover the cellular identity of individual DNs from population recordings. Notably, our efforts were facilitated by DNx01's unusual bilaterally asymmetric functional properties and large caliber axons. Taking this route to identify another, arbitrary DN would likely be more challenging and may require additional tools. For example, after functional imaging of DN populations one might focally photoactivate and image GFP within individual DN axons of interest (*Datta et al., 2008*; *Ruta et al., 2010*). Resulting morphological images could then be compared with DNs reconstructed in connectomics datasets (*Zheng et al., 2018*; *Phelps et al., 2021*).

Our work sets the stage for a more comprehensive, multiscale investigation of how the brain regulates complex limb-dependent motor behaviors. Nevertheless, overcoming several technical limitations in our study should also be a focus of future work. First, although we could achieve precise limb kinematic measurements at 100 Hz, it will be critical to record neural data at equally high temporal resolution. The fly can walk with stride frequencies of up to 20 Hz (*Mendes et al., 2013*) and leg movements during grooming occur at up to 7 Hz(*Ravbar et al., 2021*). This currently makes it difficult to relate neural activity—read out by the relatively slow fluorescence fluctuations of a genetically encoded calcium indicator—to individual joint angles and limb positions. To address this challenge, one might use faster indicators of neural activity (e.g., newer variants of GCaMP [*Zhang, 2020*] or voltage indicators [*Piatkevich et al., 2018*]). Additionally, coronal section two-photon imaging in the thoracic cervical connective with a piezo-driven objective lens limited our neural data acquisition to ~16 Hz. One might perform data acquisition at a higher rate using more advanced imaging methods including single-objective light-sheet microscopy (*Voleti et al., 2019*). Alternatively, the fly could be forced to generate slower limb movements using a motorized treadmill (*Aimon et al., 2022*). Another challenge is that DNs from the SEZ are absent in our driver line. The SEZ is considered a center for action section (*Mann et al., 2013*; *Tastekin et al., 2015*) and is known to house numerous DNs (*Namiki et al., 2018*; *Hsu and Bhandawat, 2016*). Thus, complementing our driver line with an SEZ-expressing transgene (*Simpson, 2016*) would enable a fully comprehensive recording of DN population dynamics. Nevertheless, we expect our observation to remain intact: locomotion is regulated by large DN populations in a distributed manner, while more stereotyped grooming behaviors engage fewer DNs. This would suggest a dichotomy in DN population control for flexible versus stereotyped motor behaviors. Future studies may test whether this holds true as well for wing-dependent behaviors like continuous steering during flight (*Schnell et al., 2017*; *Namiki et al., 2022*) versus stereotyped wing displays during courtship (*Pavlou and Goodwin, 2013*).

# Materials and methods

## Key resources table

| Reagent type (species) or resource | Designation | Source or reference | Identifiers | Additional information |
|---|---|---|---|---|
| Genetic reagent (*Drosophila melanogaster*) | $W; \dfrac{tub>stop>GAL80}{(CyO)}; \dfrac{MKRS}{TM3, Sb(21-21)}$ | Asahina lab (Salk Institute, San Diego, CA) (*Asahina et al., 2014*) | | |
| Genetic reagent (*D. melanogaster*) | $W; \dfrac{Otd-nls:FLP\,(attP40)}{(CyO)}; \dfrac{TM2}{TM6B(26-27)}$ | Asahina lab (Salk Institute, San Diego, CA) (*Asahina et al., 2014*) | | |
| Genetic reagent (*D. melanogaster*) | w; tubP-(FRT.GAL80); | Bloomington *Drosophila* Stock Center | BDSC: #62103 | |
| Genetic reagent (*D. melanogaster*) | w; +; R57C10-GAL4; | Bloomington *Drosophila* Stock Center (*Jenett et al., 2012*) | BDSC: #39171 | |
| Genetic reagent (*D. melanogaster*) | w; +; 10xUAS-IVS-myr::smGFP-FLAG (attP2) | Bloomington *Drosophila* Stock Center (*Nern et al., 2015*) | BDSC: #62147 | |
| Genetic reagent (*D. melanogaster*) | +[HCS]; P{20XUAS-IVSGCaMP6s} attP40; P{w[+mC]=UAS-tdTom.S}3 | Dickinson lab (Caltech, Pasadena, CA) | | |
| Genetic reagent (*D. melanogaster*) | ;P20XUAS-IVS-Syn21- OpGCaMP6f-464 p10 su(Hw)attp5; Pw[+mC]=UAS-tdTom.S3 | Dickinson lab (Caltech, Pasadena, CA) | | |
| Genetic reagent (*D. melanogaster*) | R57C10-446 Flp2::PEST in su(Hw)attP8;; HA-V5-FLAG | Bloomington *Drosophila* Stock Center (*Nern et al., 2015*) | BDSC: #64089 | |
| Genetic reagent (*D. melanogaster*) | w;;P{w[+mC]=20xUAS-DSCP>H2A::sfGFP-T2A-mKOk::Caax} JK66B | McCabe lab (EPFL, Lausanne, Switzerland) (*Jiao et al., 2022*) | | |
| Genetic reagent (*D. melanogaster*) | ;Otd-nls:FLPo (attP40)/(CyO); R57C10-GAL4, tub>GAL80>/(TM6B) | This paper | | |
| Antibody | Anti-GFP (rabbit monoclonal) | Thermo Fisher | AB2536526 | 1:500 |
| Antibody | Anti-Bruchpilot (mouse monoclonal) | Developmental Studies Hybridoma Bank | AB2314866 | 1:20 |
| Antibody | Anti-rabbit conjugated with Alexa 488 (goat polyclonal) | Thermo Fisher | AB143165 | 1:500 |
| Antibody | Anti-mouse conjugated with Alexa 633 (goat polyclonal) | Thermo Fisher | AB2535719 | 1:500 |
| Antibody | Anti-HA-tag (rabbit monoclonal) | Cell Signaling Technology | AB1549585 | 1:300 |
| Antibody | Anti-FLAG-tag DYKDDDDK (rat monoclonal) | Novus | AB1625981 | 1:150 |
| Software, algorithm | noise2void | *Krull et al., 2019* | | Denoising of red channel |
| Software, algorithm | deepinterpolation | *Lecoq et al., 2021* | | Denoising of green channel |
| Software, algorithm | DeepFly3D | *Günel et al., 2019* | | Pose estimation |
| Software, algorithm | NeuroMechFly | *Lobato-Rios et al., 2022* | | Kinematic replay, collision detection |
| Software, algorithm | utils2p | This paper | | Preprocessing of two-photon images, synchronization |
| Software, algorithm | utils video | This paper | | Creation of videos |

## Fly husbandry and stocks

All experiments were performed on female *D. melanogaster* raised at 25°C and 50% humidity on a 12 hr light–dark cycle. Flies were 10 days post-eclosion (dpe) for experiments and had been starved overnight on a wet precision wipe (Kimtech Science, 05511, USA). Sources of each genotype used are indicated in Key Resources Table.

## Olfactometer

Mass flow controllers (MFCs) were used to regulate air flow (Vögtlin, GSC-B4SA-BB23 [$2\,\mathrm{L\,min^{-1}}$], GSC-A3KA-BB22 [$100\,\mathrm{mL\,min^{-1}}$]). The larger MFC, set to $42\,\mathrm{mL\,min^{-1}}$, was used to continuously bubble odor vials, maintaining a constant head space odorant concentration. The smaller MFC was used to stimulate the fly at $41\,\mathrm{mL\,min^{-1}}$. We directed air flow using six solenoid valves (SMC, S070C-6AG-32, Japan) controlled by an Arduino Mega (Arduino, A000067, Italy). One valve in front of each of the three odor vials was used to switch between inputs from each MFC. A second valve after each odor vial was used to direct air flow either toward the fly or into an exhaust. The second valve was placed as close to the fly as possible (~10 cm) to minimize the delay between solenoid switching and odor stimulation. To direct air flow to the fly's antennae, we used a glass capillary held in place by a Sensapex zero-drift micro-manipulator (Sensapex, uMp-3, Finland). We minimized mechanical perturbations by blowing humidified (nonodorized) air onto the fly between odor trials. We used a PID (Aurora Scientific, miniPID, Canada) and visual assessment of animal behavior to confirm that switching generated minimal mechanical perturbations.

## Two-photon microscopy

We performed neural recordings using a ThorLabs Bergamo two-photon microscope (Thorlabs, Bergamo II, USA) connected to a Mai Tai DeepSee laser (Spectra Physics, Mai Tai DeepSee, USA). To perform coronal section imaging, we operated the microscope in kymograph mode using the galvo-resonant light path. Images were acquired at a magnification of 7.4× resulting in a 82.3 µm wide FOV. To prevent the cervical connective from leaving the FOV, the full range of a 100 µm piezo collar was used to scan the objective lens (Olympus XLUMPlanFLN 20× , $1.0\,\mathrm{NA}$ with $2\,\mathrm{mm}$ working distance) in the z-axis. We could achieve a frame rate of approximately $16\,\mathrm{Hz}$ by sampling 736 × 480 pixel (x- and z-axes, respectively) images and by enabling bidirectional scanning, with only one fly-back frame.

## Neural recordings

### Descending neuron population recordings

Flies were dissected to obtain optical access to the thoracic cervical connective, as described in *Chen et al., 2018*. Briefly, we opened the cuticle using a syringe and waited for the flight muscles to degrade before resecting trachea, the proventriculus, and the salivary glands. After removing these tissues covering the VNC, an implant (*Hermans et al., 2021*) was inserted into the thoracic cavity to prevent inflation of the trachea and ensure a clear view of the cervical connective for extended periods of time. Flies were then given several minutes to adapt to positioning over a spherical treadmill in the two-photon microscope system. During this adaptation period, the nozzle of the olfactometer was positioned approximately $2\,\mathrm{mm}$ in front of the fly's head. As well, the thoracic cervical connective was brought into the imaging FOV. Following alignment, their position was further adjusted to maximize fluorescence signal and minimize the possibility of neurites leaving the imaging FOV. For each fly, a minimum of five $9\,\mathrm{min}$ trials were recorded using a laser power of $9.25\,\mathrm{mW}$ at $930\,\mathrm{nm}$.

### Sparse DNx01 recordings

Recordings of DNx01s in the R65D11-GAL4 driver line were performed as described in *Chen, 2022*. This differed only slightly from DN population recordings in the following ways. First, flies were not starved. Second, the Gal4 drove expression of OpGCaMP6f rather than GCaMP6s. Third, a thoracic implant was not used. Fourth, instead of being presented with a constant flow of air and odors, animals were stimulated with $CO_2$ puffs of alternating length ($0.5\,\mathrm{s}$, $2\,\mathrm{s}$) spaced $40\,\mathrm{s}$ apart. A higher pixel dwell time was used to achieve acceptably high imaging signal-to-noise. This resulted in a

slower two-photon image acquisition rate (4.3 fps), which was matched by a slower behavior recording frequency (30 Hz). For additional details and a description of the stimulation system, see *Chen, 2022*. Note that images of the connective in *Figure 5a and e* appear to have different heights due to a difference in z-step size during image acquisition.

## Postprocessing of two-photon imaging data

### Descending neuron population recordings

Binary output files from ThorImage (Thorlabs, ThorImage 3.2, USA) were converted into separate tiff files for each channel using custom Python code (https://doi.org/10.5281/zenodo.5501119). Images acquired from the red channel were then denoised (*Krull et al., 2019*) and two-way alignment offset was corrected (https://doi.org/10.5281/zenodo.6475468) based on denoised images. We then used optic flow estimation to correct image translations and deformations based on the denoised red channel images (https://doi.org/10.5281/zenodo.6475525). The green channel was then corrected based on the motion field estimated from the red channel. Finally, we trained a DeepInterpolation network (*Lecoq et al., 2021*) for each fly using the first 500 motion-corrected green channel images from each experimental trial (batch size = 20; epochs = 20; pre-post frames = 30). The first and the last trials were used as validation datasets. The trained network was then used to denoise green channel images.

Baseline fluorescence was then determined on a fly-wise and pixel-wise basis across all trials. The baseline of a pixel was defined as the minimum 'mean of 15 consecutive values' across all experimental trials. Motion correction introduces zeroes to two-photon images in background regions that were out of the FOV prior to warping. Therefore, values close to zero (floating point inaccuracy) were set to the maximum of the datatype of the array. This means essentially ignoring these pixels and their immediate surroundings for baseline computations. We used the baseline image $F_0$ to calculate $\%\frac{F-F_0}{F_0}$ images. These images were only used for visualization. To identify ROIs/neurons, we created a maximum intensity projection of $\%\frac{F-F_0}{F_0}$ images and manually annotated ROIs. The $\%\frac{F-F_0}{F_0}$ of each ROI was computed by first spatially averaging its raw pixel values. We then calculated the baseline of this average as described for a single pixel above. For brevity and readability, we refer to $\%\frac{F-F_0}{F_0}$ as $\%\Delta F/F$ throughout the article.

### Sparse DNx01 descending neuron recordings

ROIs were detected using AxoID. Raw, non-denoised traces were used for analysis. For more details concerning data postprocessing, see *Chen, 2022*.

## Behavior classification and quantification

### Behavior measurement system

The behavior of tethered animals was recorded using a previously described (*Günel et al., 2019*) 7-camera (Basler, acA1920-150um, Germany) system. Animals were illuminated using an infrared (850 nm) ring light (CSS, LDR2-74IR2-850-LA, Japan). To track the joint positions of each leg, six cameras were equipped with 94 mm focal length 1.00× InfiniStix lenses (Infinity, 94 mm/1.00×, USA). All cameras recorded data at 100 fps and were synchronized using a hardware trigger. For more details, see *Günel et al., 2019*.

### Inferring fictive locomotor trajectories

Video data from the front camera were processed using FicTrac (*Moore et al., 2014*) to track ball movements. This camera was outfitted with a lens allowing adjustable focus and zoom (Computar, MLM3X-MP, 0.3X-1X, 1:4.5, Japan). To improve tracking accuracy, the quality factor of FicTrac was set to 40. The circumference of the ball was detected automatically using a Hough circle transform on the mean projection of all images for a given experimental trial. To determine the vertical angular FOV, $\alpha$, the value given in the specifications of the Computar lens (8.74°) had to be adjusted, accommodating a smaller sensor size. We first determined the focal length to be 43.18 mm using *Equation 1*, where $H$ is the height of the sensor. This was set to 6.6 mm for a 2/3″ sensor.

$$f = \frac{H}{2tan(\frac{\alpha}{2})} \tag{1}$$

The ROI of the Basler camera was set to 960 × 480 pixels, reducing the effective sensor height from $5.8\,mm$ to $2.32\,mm$. Rearranging *Equation 1* and plugging in $f$ and $H$ yields a vertical angular FOV of 3.05°. Since the camera was already aligned with the animal, the camera-to-animal transform was set to zero. To obtain the fly's trajectory, we developed custom code that integrates rotational velocities (https://github.com/NeLy-EPFL/utils_ballrot; *Aymanns, 2022*; copy archived at swh:1:rev:7247f448be62f349bb528ad70633b4b41be5bbaf).

### Postprocessing of 3D pose estimates

Outliers in 3D pose data were corrected as described in *Chen, 2022*. Briefly, we detected outliers based on changes in leg segment length and chose the pair of cameras with minimal reprojection error for triangulation. After outlier correction, the data were aligned and joint angles were computed using published code (*Lobato-Rios et al., 2022*): https://github.com/NeLy-EPFL/df3dPostProcessing/tree/outlier_correction.

### Classification of behaviors

Behaviors were classified based on limb joint angles using the approach described in *Whiteway et al., 2021*. Briefly, a network was trained using 1 min of annotations for each fly and heuristic labels. Motion energy, ball rotations, and joint positions were used to generate the heuristic labels. To compute the motion energy, each joint position was convolved with the finite difference coefficients of length nine, estimating the first derivative. After computing the $L^1$-norm, the signal was filtered with a tenth-order low-pass Butterworth filter of critical frequency $4\,Hz$. The total, front, and hind motion energy were computed by summing over all joints, all front leg joints, and all hindleg joints, respectively. Forward ball rotation speeds were processed using the same Butterworth filter described above. First, we assigned heuristic labels for walking by thresholding the filtered forward walking velocity at $0.5\,mm\,s^{-1}$. The remaining frames with a motion energy smaller than 0.3 were then classified as resting. Next, heuristic labels for front movements (front motion energy > 0.2 and hind motion energy < 0.2) and posterior movements (front motion energy < 0.2 and hind motion energy > 0.2) were assigned to all remaining frames. The front movement labels were further split into head grooming and front leg rubbing by thresholding the height of the front leg tarsi (the average between left and the right tarsi) at 0.05. After each step, a hysteresis filter was applied. This filter only changes state when at least 50 consecutive frames are in a new state. Based on a hyperparameter search using 'leave one fly out' cross-validation on the hand labels (*Figure 1—figure supplement 1g*), we selected the weights of $\lambda_{pred} = 0$ and $\lambda_{weak} = 1$ for the loss.

### Biomechanical simulation and antennal collision detection

To infer limb-antennal collisions, we performed kinematic replay using the NeuroMechFly physics simulation framework as described in *Lobato-Rios et al., 2022*. We used joint angles to replay real limb movements in the simulation. To avoid model explosion and accumulating errors over long simulation times, we ran kinematic replay on time segments shorter than the full experimental trials. These segments consisted of individual head grooming and front leg rubbing events. The default head angle was fixed to 20°. The aristae yaw values were set to –32° and 32° and pedicel yaw values were set to –33° and 33° for the left and right sides, respectively. To speed up the simulation, we only detected collisions between the front legs and head segments.

## Confocal imaging of the brain and ventral nerve cord

Confocal images were acquired using a Zeiss LSM700 microscope. These images were then registered to a brain and VNC template described in *Chen, 2022* using the method from *Jefferis et al., 2007*. Brain and VNC sample preparation was performed as described in *Chen, 2022*. Both primary and secondary antibodies were applied for 24 hr and the sample was rinsed 2–3 times after each step. Antibodies and concentrations used for staining are indicated in Key Resources Table.

## Electron microscopy identification and tracing

Within an electron microscopy dataset of the VNC and neck connective (*Phelps et al., 2021*), we identified the pair of DNx01s based on their large-caliber axons in the cervical connective positioned ventral to the giant fiber neurons axons (*Figure 5I*). We then manually reconstructed all branches of one DNx01 neuron using CATMAID (*Saalfeld et al., 2009*; *Schneider-Mizell et al., 2016*) as described in *Phelps et al., 2021*. The reconstructed neuron was then registered to the female VNC standard template (*Bogovic et al., 2020*) using an elastix-based atlas registration as described in *Phelps et al., 2021*.

## Data analysis

Animals were excluded from analysis if they produced irregular behavior or fewer than 75 neuronal ROIs could be manually identified in two-photon imaging data.

### Linear regression modeling

We relied primarily on regression techniques to link behavioral and neural data. Here, we first describe the general approaches used and then discuss the details and modifications for individual figure panels. To evaluate the success of regression models, we calculated explained variance in the form of the coefficient of determination ($R^2$) and unique explained variance (UEV) (*Musall et al., 2019*). The explained variance is a measure of how much additional variance is explained by the model compared to an intercept-only model (i.e., approximating the data by taking the mean). A definition of the coefficient of determination can be found in *Equation 2*, where SSE is the sum of squares of the error, SST is the total sum of squares, $y_i$ is a data point, $\hat{y}_i$ is the prediction of $y_i$, and $\bar{y}$ is the mean of all data points.

$$R^2 = 1 - \frac{\text{SSE}}{\text{SST}} = 1 - \frac{\sum_i (y_i - \hat{y}_i)^2}{\sum_i (y_i - \bar{y})^2} \tag{2}$$

Note that $R^2$ becomes negative when SSE is larger than SST. This is the case when the model prediction introduces additional variance due to overfitting. UEV is a measure for the importance of individual or a subset of regressors. It is computed as the reduction in $R^2$ when a particular subset of regressors is randomly shuffled (*Equation 3*).

$$\text{UEV} = R^2_{\text{intact}} - R^2_{\text{shuffled}} \tag{3}$$

We performed fivefold cross-validation for all of our regression results to ensure good generalization. Nonregularized linear regression sometimes led to overfitting and negative $R^2$ values. Therefore, we used ridge regression. The ridge coefficient was determined using fivefold nested cross-validation on the training data set. In some cases, we still observed small negative $R^2$ values after applying ridge regularization. These were set to zero, in particular to avoid problems when computing UEVs. To account for the long decay dynamics of our calcium indicator, we convolved behavior variables with an approximation of the crf (*Equation 4*).

$$\text{crf}(t) = -e^{-at} + e^{-bt} \tag{4}$$

We used $a = 7.4$ and $b = 0.3$ to approximate the rise and decay times, respectively, as reported in *Chen et al., 2013*. We also normalized the function to integrate to one on the interval $0 \, 30 \, \text{s}$ In *Figure 2* and *Figure 2—figure supplement 2*, we predicted behavior from neural activity. To accomplish this, we trained models for all pairs of behaviors and neurons (e.g., walking and ROI 41 for the upper plot of *Figure 2a*). The target variable is a binary variable indicating whether the fly is walking or not. This was convolved with the crf (black line in *Figure 2a*). The single regressor besides the intercept in the model is the $\Delta F/F$ of a single neuron. *Figure 2b* and *Figure 2—figure supplement 2c* show the $R^2$ values for all models. Each neuron was then assigned to be encoding the behavior with the maximum $R^2$. For neurons with maxima smaller than 5%, no behavior was assigned. In *Figure 2—figure supplement 1*, using the approach described in the previous section, we observed that some neurons appear to predict both walking and posterior movements, while others predict both head grooming and front leg rubbing. Because fluorescence decays slowly following the cessation of neural activity, this may be caused by the frequent sequential occurrence of two behaviors. For instance, if front leg rubbing often occurs after head grooming, the $\Delta F/F$ of a head groom

encoding neuron may still be elevated during front leg rubbing. This can lead to false positives in our analysis. To address this potential artifact, we predicted neural activity from multiple behavior regressors (i.e., binary behavior indicators convolved with the crf). We then calculated the UEV for each behavior regressor. For example, when the front leg rubbing regressor is shuffled the $R^2$ will not decrease by much because the head grooming regressor includes the expected decay through convolution with a crf. The model of *Figure 1—figure supplement 1a* includes a walking and a posterior movements regressor. For *Figure 2—figure supplement 1b*, the model includes a head grooming and a front leg rubbing regressor. No other regressors were included in these models and for a given model, only data during one of these two behaviors were used. In *Figure 3* and *Figure 3—figure supplement 2*, for *Figure 3a*, we used all behavior regressors and the $\Delta F/F$ of all neurons to predict ball rotation speeds convolved with a crf. We then temporally shuffled each of the neural regressors to calculate their UEVs. Since knowing whether the fly is walking or not provides some information about forward speed, the $R^2_{\text{intact}}$ did not decrease to zero in the speed prediction (*Figure 3a*, second row). We address this issue in *Figure 3b* by only including walking frames but otherwise using the same approach as in *Figure 3a*. To pinpoint the encoding of ball rotations to individual neurons (*Figure 3c*), we made two changes to our approach. First, instead of predicting turning in general, we split the turning velocity into right and left turning. Second, we only included the $\Delta F/F$ of a single neuron in the model rather than the data from all neurons. As before, the target variables were spherical treadmill rotation speeds—right turning, left turning, and forward walking—convolved with the crf. In *Figure 4* and *Figure 4—figure supplement 1*, to model odor encoding, we predicted the activity of a single neuron using behavior regressors, including crf-convolved spherical treadmill rotation speeds, and odor regressors constructed by convolving the crf with a binary variable indicating whether a given odor was present or not. To determine how much neural variance can be explained by the behavior regressors in our model, we shuffled both odor regressors (*Figure 4b*, top row). We then calculated the UEV for each odor by first computing the $R^2$ of the model with all regressors intact, and then subtracting the $R^2$ of the model after shuffling the odor regressor in question (*Figure 4b*, bottom). In *Figure 4c and d*, the intact model's prediction (blue) and the shuffled model's prediction (red) are shown. In *Figure 4—figure supplement 2*, to see whether a neuron's behavior encoding could change depending on the context of which odor is present, we only examined the most commonly encoded behaviors: walking and head grooming. Each row is equivalent to the walking or head grooming rows in *Figure 2—figure supplement 2b*. However, here we only used subsets of the data to train and validate our models. Each row only uses data when one odor (or humidified air) is present. If there is no context dependence, each of the three rows should look identical. However, we note that due to the significant reduction in the amount of data for each model, noise can introduce variation across conditions. Humidified air data was subsampled to match the amount of data available for the odors MSC and ACV. To test whether the encoding was significantly different across contexts, we used a two-sided Mann–Whitney $U$-test. The data points are individual cross-validation folds from each trial. Elsewhere in this study we report cross-validation means. In *Figure 5b*, we perform a regression to predict one neuron's activity using the intercept as well as a crf-convolved antennal collision regressor derived from data in *Figure 5d*.

## Kernel density estimation

We performed 2D kernel density estimation (*Figure 2h and i*) using SciPy's gaussian_kde (*Virtanen et al., 2020*). We performed 1D kernel density estimation (*Figure 3h and j*, *Figure 3—figure supplement 2d and f*) using sklearn (*Pedregosa, 2011*). We normalized kernel density estimates to correct for the variable density of neurons across the connective. The normalization factor was computed as a kernel density estimate of all annotated neuron locations. The kernel bandwidth was determined using leave-one-out cross-validation. This maximizes the log-likelihood of each sample under the model. Data from all flies were used to determine the bandwidth.

## Principal component analysis

We performed PCA on DN population data. First, as in *Kato et al., 2015*, we observed that PCs from time derivatives of fluorescence traces produce more organized state space trajectories. Therefore, we calculated the derivative of the $\Delta F/F$ traces for each neuron (extracted from nondenoised imaging

data) using total variation regularization (*Chartrand, 2011*). Empirically, we found that a regularization parameter of 5000 strikes a good balance between bias and variance in the derivatives. We then performed PCA on the derivatives of all neural data during walking only. This allowed us to specifically ask whether, during walking, neural activity largely remained constant or diverged as a function of specific locomotor features. We then embedded walking and resting neural data epochs into the same PC space (i.e., we did not refit the PCs for resting). We visualized the loadings of individual ROIs/neurons using vectors to illustrate how these neurons influence the position in PC space. For clarity, we only show vectors for whom $w_{pc1}^2 + w_{pc2}^2 > 0.1$, where $w_{pc1}$ and $w_{pc2}$ are the loadings of the first and second principal components.

## Linear discriminant analysis

We trained linear discriminant models to distinguish between ACV and humidified air, as well as MSC and humidified air (*Figure 4a*). To evaluate classification accuracy, the classification score was cross-validated. Input data to the model were either behavior regressors or neural residuals. The residuals were computed using ridge regression (as described above) with behavior regressors as input. In both cases, the behavior regressors were convolved with the crf.

## Event-triggered averaging

Here, 'events' describe individual epochs of a particular behavior. To perform event-triggered averaging of images/videos (*Videos 2–4*), we first identified all events that had no similar event in the previous 1 s. Raw microscope recordings were then chopped into blocks starting 1 s prior to event onset and lasting 4 s after event onset or until the end of the event, whichever was shorter. Each block was then converted into $\Delta F/F$ using the mean of the first five frames as a baseline. All blocks were then temporally aligned and averaged frame-by-frame. We discarded behavior videos with fewer than 50 events 1 s after event onset. Event-triggered averaging of neural traces (*Figure 5g and h*) was performed in a similar fashion. However, instead of using raw images, $\Delta F/F$ traces were used and no block-wise $\Delta F/F$ was computed. The values were then averaged one time point at a time.

## Correlation coefficients

We calculated the Pearson's correlation coefficient for each trial between the raw $\Delta F/F$ trace and a time-shifted $\Delta F/F$ trace calculated using denoised images (*Figure 1—figure supplement 1b*). We then either grouped the values by time lag and averaged each group (bottom), or we found the time shift with maximal cross-correlation. In *Figure 3d* and *Figure 3—figure supplement 2b*, we perform several types of correlation analyses. First, the larger matrices show the correlation between neurons. This is computed as the Pearson's correlation coefficient between the $\Delta F/F$ values of a pair of neurons. Correlations between neural activity and turning or walking speed were calculated as the Pearson's correlation coefficient between the $\Delta F/F$ values of a neuron and corresponding spherical treadmill rotations using only data when the fly was classified as walking. All of the above calculations were performed on individual trials and then averaged across all trials.

## Hierarchical clustering

We used Ward's method (*Ward, 1963*) to hierarchically cluster and sort neurons based on their correlation (*Figure 3d*). The distance between pairs of neurons was set to $1 - r$, where $r$ is the Pearson's correlation coefficient for the two neurons.

# Acknowledgements

We thank K Asahina (Salk Institute, San Diego, USA) and B McCabe (EPFL, Lausanne, Switzerland) for transgenic *Drosophila* strains. We thank J Phelps for discussions and assistance with manual annotation of the EM dataset. PR acknowledges support from an SNSF Project Grant (175667) and an

SNSF Eccellenza Grant (181239). FA acknowledges support from a Boehringer Ingelheim Fonds PhD stipend.

## Additional information

### Funding

| Funder | Grant reference number | Author |
| --- | --- | --- |
| Boehringer Ingelheim Fonds | | Florian Aymanns |
| Schweizerischer Nationalfonds zur Förderung der Wissenschaftlichen Forschung | 175667 | Pavan Ramdya |
| Schweizerischer Nationalfonds zur Förderung der Wissenschaftlichen Forschung | 181239 | Pavan Ramdya |

The funders had no role in study design, data collection and interpretation, or the decision to submit the work for publication.

### Author contributions

Florian Aymanns, Data curation, Software, Formal analysis, Validation, Investigation, Visualization, Methodology, Writing - original draft, Writing - review and editing; Chin-Lin Chen, Data curation, Formal analysis, Validation, Investigation, Visualization, Writing - review and editing; Pavan Ramdya, Conceptualization, Resources, Supervision, Funding acquisition, Methodology, Writing - original draft, Project administration, Writing - review and editing

### Author ORCIDs

Florian Aymanns ⓘ http://orcid.org/0000-0003-4290-7244
Chin-Lin Chen ⓘ http://orcid.org/0000-0002-4968-4920
Pavan Ramdya ⓘ http://orcid.org/0000-0001-5425-4610

### Decision letter and Author response

Decision letter https://doi.org/10.7554/eLife.81527.sa1
Author response https://doi.org/10.7554/eLife.81527.sa2

## Additional files

### Supplementary files
• MDAR checklist

### Data availability

Data are available at: https://dataverse.harvard.edu/dataverse/DNs. Analysis code is available at: https://github.com/NeLy-EPFL/DN_population_analysis (copy archived at swh:1:rev:7c5527dae0d6ff760ddda657d3194cc19ccda3eb).

The following datasets were generated:

| Author(s) | Year | Dataset title | Dataset URL | Database and Identifier |
| --- | --- | --- | --- | --- |
| Aymanns F, Chen C-L, Ramdya P | 2022 | R65D11 two-photon recording data | https://doi.org/10.7910/DVN/YU1N1A | Harvard Dataverse, 10.7910/DVN/YU1N1A |
| Aymanns F, Chen C-L, Ramdya P | 2022 | Brain_only_GAL4_population_data | https://doi.org/10.7910/DVN/QQMNQK | Harvard Dataverse, 10.7910/DVN/QQMNQK |

| Author(s) | Year | Dataset title | Dataset URL | Database and Identifier |
|-----------|------|---------------|-------------|-------------------------|
| Aymanns F, Chen C-L, Ramdya P | 2022 | Confocal images | https://doi.org/10.7910/DVN/KTQT27 | Harvard Dataverse, 10.7910/DVN/KTQT27 |

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
