## [Editor Report]

This article uses a genetically encoded calcium indicator to assess neural activity across a population of axons connecting the fly’s brain to its ventral nerve cord while the tethered fly behaves on a floating ball. The preparation and large-scale analysis represent a significant step forward in determining how the brain compresses sensory and state information to convey commands to the ventral nervous system for behavior execution by motor circuits.

---

## [Decision Letter]

**Decision letter after peer review:**

Thank you for submitting your article "Descending neuron population dynamics during odor-evoked and spontaneous limb-dependent behaviors" for consideration by *eLife*. Your article has been reviewed by 2 peer reviewers, and the evaluation has been overseen by a Reviewing Editor and K VijayRaghavan as the Senior Editor. The reviewers have opted to remain anonymous.

Essential revisions:

The authors use a challenging preparation in which they were able to record from up to 100 DNs simultaneously in tethered flies performing spontaneous and odor-evoked behaviors on a treadmill. They combine their recordings with motion capture approaches and automated behavioral classification, which allows them to correlate DN population activity with behavior on an unprecedented scale. This approach is valuable and adds a different perspective to previously published studies aimed at tying individual, often command-like DNs to different behaviors and characterizing their activity in detail. The authors use relatively complex analysis methods, which is necessary due to the rich behavioral and neuronal activity data sets. In several instances, they verify that their conclusions drawn from the output of their analysis pipeline hold when tested with more simple and common analysis methods (see for example Figures 5g or 4c). After correlating the activity of the DN population with aspects of walking and grooming, they outline an approach that allowed them to identify a single pair of DNs from the population data set. Overall, the study combines several cutting-edge methods and significantly adds to our understanding of descending motor control. For example, the authors demonstrate that a large number of DNs likely contributes to turning, whereas changes in walking speed are encoded, perhaps driven, by fewer, more distributed DNs.

In our consultation, we agreed that the data and presentation are strong, and the biological findings are important. Some word choices are confusing, and some additional discussion of implications, comparisons, or limits is warranted, but can be achieved with text revision. It is with this in mind that the major points are to be addressed. These points are given below.

1) The use of the word "encoding" is problematic. It may be best reserved for when we know that a neuron's activity pattern occurs in direct response to specific sensory inputs or causes specific motor outputs. The changes in fluorescence seen here correlate better with certain behaviors than others, but the sensory input is not controlled, the time resolution is limited, and the causality is not shown. The experimental data is fine – but we would describe it as signal correlation rather than encoding.

We would prefer to see that word – encoding – removed. If not, an early definition and an extended Discussion of the way the authors are using the word with a clear presentation of the caveats would be acceptable.

2) How do the authors handle statistical significance in rare behaviors or comparisons of correlation strength between behaviors that occur with different frequencies? Turning doesn't happen that often. Particular combinations of joint angles might not be common. If you bin all of the micromovements that compose grooming, you'd aggregate a lot more signal than for any of the individual limb positions. Do you normalize by time? This is a concern for assessing the conclusion that the calcium signal correlates better with higher-order behavior categories (walking or grooming) than it does with shorter, rarer movements or limb positions.

3) In their experiments, the authors did not perturb the activity of any of the DNs, either by activation or silencing. Moreover, the temporal resolution of the DN population recordings is relatively low compared to, for example, single cell patch-clamp recordings. This is fair given the scope of the study, but as a consequence it remains unclear whether a DN whose activity is correlated with a certain behavior is driving this particular behavior, or whether the DN is activated because the behavior is executed. The latter could for example be due to sensory feedback. This caveat makes it challenging to interpret the results presented since a causal link between DN activity and behavior cannot be assumed. Overall, the authors are relatively careful when interpreting their data, but there are several instances where they overinterpret their findings. These instances need to be addressed and clarified:

– The statements in line 63ff regarding the reasoning for using an approach that allows parallel recordings from many DNs do not seem ideal for several reasons:

a) To resolve how different DNs modulate ongoing behavior, it seems the best approach would be to activate these DNs individually or in groups to get an idea of their behavioral effects and establish causality, as opposed to correlating their activity with spontaneous behavior at low temporal resolution.

b) In order to establish how DNs are recruited depending on sensory context, it would seem more important to provide a large variety of sensory contexts rather than recording from many neurons at the same time. It would be perfectly fine to establish sensory context for one DN at a time.

c) If one main goal was to establish whether DNs provide raw sensory information or processed, abstract commands, it would seem more important to have precise control over sensory stimuli and perhaps a higher temporal resolution on the DN activity readout rather than recording from multiple DNs in spontaneously behaving flies at relatively low temporal resolution.

The approach used by the authors is very valuable and it provides insights that single neuron recordings or optogenetic activation will never be able to deliver, but the reasons stated in the introduction do not really highlight the strengths of the present study.

4) – Line 82ff: The experiments presented do not rule out a strong context-dependence of DN activity in general. They merely show that many of the DNs found to 'encode' aspects of walking and grooming do so independently of whether the behavior was spontaneous or facilitated by olfactory stimulation. However, it is absolutely conceivable that different subsets of DNs control turning when it is induced by visual vs. mechanosensory vs. unilateral olfactory cues, for example. As far as I can see, this possibility has not been explored or tested in any of the experiments presented.

5) – l. 89: 'global view' seems overstated given that the authors recorded from <100 out of about 1000 DNs (in one species). It is certainly a wider view than we had before!

6) – In l. 167, the authors suggest that DNs encode high-level behaviors and in l. 136f they speculate that DNs likely drive these behaviors. This would seem like a reasonable assumption for descending neurons. However, when the authors follow up on one of the DNs they identified individually using EM tracing and a sparse driver lines, they actually show that this particular DN (DNx01) neither encodes high-level behaviors, nor does it seem to drive the behavior its activity is most strongly correlated with (head grooming). Instead, DNx01 seems to convey simple, mechanosensory inputs from the antennae to the VNC (figure 5g). How do the authors reconcile this observation with the general underlying assumption that the large majority of DNs they recorded drive behavior rather than encode sensory feedback?

7) – It seems that it was possible to identify the DNx01s due to their strong sensory responses and large axons that were easily distinguishable in EM stacks and functional imaging. It would be nice if the authors could discuss a little further whether and how it will be feasible to expand this approach to other DNs in the future.

8) In l. 254, the authors suggest that turning might be driven by asymmetries in VNC networks rather than by asymmetric activation of VNC networks via DNs. This model is hard to reconcile with existing knowledge about motor control. It is possible for VNC networks to independently generate asymmetric activity of course (for example in response to unilateral local sensory inputs). However, if DNs are not asymmetrically activated to drive turning, how would the brain be able to drive voluntary turns? What is the underlying model?

9) The authors use NeuroMechFly, a biomechanical simulation of the *Drosophila* body, to play back movements recorded by their motion capturing pipeline and detect collisions between the front legs and the right and left antennae. What is the reason for using such an indirect approach to detect potential antennal deflections? It seems the authors should be able to detect antennal deflections unambiguously in their video recordings. From looking at the supplemental videos, the leg movements of the model and the fly do not always seem to match perfectly (as would be expected from an approximation). Did the authors verify that their predictions were accurate?

10) The definition of 'posterior movement' (l.127) is vague. Does this include every instance of abdominal bending? Were hind leg movements and abdominal bending treated the same way in the analysis? Why would that be a reasonable simplification? It would be nice if the authors could expand on this a little bit.

11) The Discussion of the implications should be expanded, in particular, to include parallels to the ascending neurons, whose activity also seems to correlate with higher order/larger scale representations. An explicit comparison to electrophysiological recordings of DNs should be included. An advantage of calcium imaging over electrophysiological recordings is the population aspect – signal can be compared among neurons to determine patterns and co-activation. How was this employed here? Why express GCaMP in most DNs at once, rather than in specific DN split GAL4 lines? Whether fluorescence changes correlated with behaviors could have been explored in both cases.

---

## [Author Response]

Essential revisions:The authors use a challenging preparation in which they were able to record from up to 100 DNs simultaneously in tethered flies performing spontaneous and odor-evoked behaviors on a treadmill. They combine their recordings with motion capture approaches and automated behavioral classification, which allows them to correlate DN population activity with behavior on an unprecedented scale. This approach is valuable and adds a different perspective to previously published studies aimed at tying individual, often command-like DNs to different behaviors and characterizing their activity in detail. The authors use relatively complex analysis methods, which is necessary due to the rich behavioral and neuronal activity data sets. In several instances, they verify that their conclusions drawn from the output of their analysis pipeline hold when tested with more simple and common analysis methods (see for example Figures 5g or 4c). After correlating the activity of the DN population with aspects of walking and grooming, they outline an approach that allowed them to identify a single pair of DNs from the population data set. Overall, the study combines several cutting-edge methods and significantly adds to our understanding of descending motor control. For example, the authors demonstrate that a large number of DNs likely contributes to turning, whereas changes in walking speed are encoded, perhaps driven, by fewer, more distributed DNs.

We thank the Editor and Reviewers for their positive appreciation of our work.

In our consultation, we agreed that the data and presentation are strong, and the biological findings are important. Some word choices are confusing, and some additional discussion of implications, comparisons, or limits is warranted, but can be achieved with text revision. It is with this in mind that the major points are to be addressed. These points are given below.1) The use of the word "encoding" is problematic. It may be best reserved for when we know that a neuron's activity pattern occurs in direct response to specific sensory inputs or causes specific motor outputs. The changes in fluorescence seen here correlate better with certain behaviors than others, but the sensory input is not controlled, the time resolution is limited, and the causality is not shown. The experimental data is fine – but we would describe it as signal correlation rather than encoding.We would prefer to see that word – encoding – removed. If not, an early definition and an extended Discussion of the way the authors are using the word with a clear presentation of the caveats would be acceptable.

We agree that this is an interesting point for discussion. Notably, other laboratories have also previously described the ‘encoding’ of insect descending neurons (e.g., in locust: J.R. Gray et al., *J. Comp Physiol A*, 2010; in *Drosophila*: M. Suver et al., *J. Neurosci.* 2016; in flies: S. Nicolas et al., *J. Neurosci.*, 2018; in stick insects: B. Jaske et al., *J. Neurophysiol.*, 2021). This word is less cumbersome than the full phrase ‘the activity of X neuron is correlated with Y’. Therefore, in one instance we have exchanged “encoding” for “is active during” but we have otherwise opted to use the word “encoding” in the manuscript. As suggested, we provide an early definition and discussion of the way we are using the word with a presentation of caveats.

2) How do the authors handle statistical significance in rare behaviors or comparisons of correlation strength between behaviors that occur with different frequencies? Turning doesn't happen that often. Particular combinations of joint angles might not be common. If you bin all of the micromovements that compose grooming, you'd aggregate a lot more signal than for any of the individual limb positions. Do you normalize by time? This is a concern for assessing the conclusion that the calcium signal correlates better with higher-order behavior categories (walking or grooming) than it does with shorter, rarer movements or limb positions.

The relative frequencies of different behaviors is an important consideration when quantifying their correlations with DN activity. We found a strong encoding of turning among many DNs (Figure 3 and Figure 3 —figure supplement 2 (previously Figure S7)) suggesting that there are sufficient examples of this behavior. In the case of more rare behaviors we addressed these concerns by balancing the data. In other words, we ensure that an equal amount of each behavior is used to calculate relative encoding. This was particularly important in the case of rare behaviors that are likely to occur after more common actions. Specifically, in Figure 2 —figure supplement 1 (previously Figure S2), we balanced the data to confirm that DNs encode walking rather than posterior movements, and that other DNs encode head grooming rather than front leg rubbing.

With respect to joint angle probabilities, in the revised manuscript we now include a new analysis (new Figure 1 —figure supplement 2) which shows that a wide range of joint angles are measured for different leg joint degrees of freedom. In particular, we find a wider distribution of FTi pitch angles compared with ThC pitch angles.

3) In their experiments, the authors did not perturb the activity of any of the DNs, either by activation or silencing. Moreover, the temporal resolution of the DN population recordings is relatively low compared to, for example, single cell patch-clamp recordings. This is fair given the scope of the study, but as a consequence it remains unclear whether a DN whose activity is correlated with a certain behavior is driving this particular behavior, or whether the DN is activated because the behavior is executed. The latter could for example be due to sensory feedback. This caveat makes it challenging to interpret the results presented since a causal link between DN activity and behavior cannot be assumed. Overall, the authors are relatively careful when interpreting their data, but there are several instances where they overinterpret their findings. These instances need to be addressed and clarified:– The statements in line 63ff regarding the reasoning for using an approach that allows parallel recordings from many DNs do not seem ideal for several reasons:a) To resolve how different DNs modulate ongoing behavior, it seems the best approach would be to activate these DNs individually or in groups to get an idea of their behavioral effects and establish causality, as opposed to correlating their activity with spontaneous behavior at low temporal resolution.

We agree with the Reviewer that to establish causality it would be best to activate groups of DNs in behaving animals (i.e., as in Cande et al., *eLife* 2018 but for multiple DN classes at once). However, this approach is currently technically challenging for several reasons. First, it is not yet clear how one might target many different arbitrary combinations of DNs using existing genetic reagents. For example, simply combining split-Gal4 driver lines can lead to off-target expression and does not permit the co-activation of more than several DN classes at once. This approach also yields a massive combinatorial space of required experiments. Our approach of measuring the population code of DNs is an important step toward motivating specific co-activation experiments in future work.

We note that a future alternative could be to perform spatially restricted (e.g., SLM-based) optogenetic activation of multiple DN axons in the neck connective of behaving animals. However, the axial resolution of SLM-based stimulation is too broad with respect to the size of DN axons. As well, experiments in behaving animals suffer from constant tissue movement and deformation. This makes precise optogenetic targeting impossible without extremely fast closed-loop stimulation control.

b) In order to establish how DNs are recruited depending on sensory context, it would seem more important to provide a large variety of sensory contexts rather than recording from many neurons at the same time. It would be perfectly fine to establish sensory context for one DN at a time.c) If one main goal was to establish whether DNs provide raw sensory information or processed, abstract commands, it would seem more important to have precise control over sensory stimuli and perhaps a higher temporal resolution on the DN activity readout rather than recording from multiple DNs in spontaneously behaving flies at relatively low temporal resolution.The approach used by the authors is very valuable and it provides insights that single neuron recordings or optogenetic activation will never be able to deliver, but the reasons stated in the introduction do not really highlight the strengths of the present study.

We agree with the Reviewer that measuring individual DNs’ precise responses to a wide panel of sensory stimuli is a valuable and complementary approach to population recordings. However, we note that the proposed investigation requires a combinatorially massive number of experiments given (i) the number of different DN classes and (ii) the vast number of sensory cues across visual, olfactory, gustatory, and mechanosensory modalities. We believe that population recordings can therefore contribute to this endeavor by enabling the more rapid investigation of sensory-based recruitment of DNs in future work. We discuss these points in the Introduction of the revised manuscript.

4) – Line 82ff: The experiments presented do not rule out a strong context-dependence of DN activity in general. They merely show that many of the DNs found to 'encode' aspects of walking and grooming do so independently of whether the behavior was spontaneous or facilitated by olfactory stimulation. However, it is absolutely conceivable that different subsets of DNs control turning when it is induced by visual vs. mechanosensory vs. unilateral olfactory cues, for example. As far as I can see, this possibility has not been explored or tested in any of the experiments presented.

We agree with the Reviewer that generalizing our findings across a wider panel of sensory stimuli is a valuable future direction. Therefore, in the revised manuscript’s Introduction we now state:

“These data suggest a lack of strong context dependence in DN population recruitment. In the future, this finding can be strengthened and made more general by examining DN population activity in the presence of other visual, mechanosensory and olfactory cues.”

5) – l. 89: 'global view' seems overstated given that the authors recorded from <100 out of about 1000 DNs (in one species). It is certainly a wider view than we had before!

We have modified the text to read “…an expansive view…”.

6) – In l. 167, the authors suggest that DNs encode high-level behaviors and in l. 136f they speculate that DNs likely drive these behaviors. This would seem like a reasonable assumption for descending neurons. However, when the authors follow up on one of the DNs they identified individually using EM tracing and a sparse driver lines, they actually show that this particular DN (DNx01) neither encodes high-level behaviors, nor does it seem to drive the behavior its activity is most strongly correlated with (head grooming). Instead, DNx01 seems to convey simple, mechanosensory inputs from the antennae to the VNC (figure 5g). How do the authors reconcile this observation with the general underlying assumption that the large majority of DNs they recorded drive behavior rather than encode sensory feedback?

We believe that DNx01 is likely exceptional in encoding sensory feedback. This is underpinned by its unusually large caliber axons. However, we do not exclude the possibility that selective DNx01 activation may drive or suppress specific behaviors (e.g., generating a stance stabilization reflex to strong gusts of wind). As we now state in the Discussion of the revised manuscript, testing this hypothesis will require the generation of a sparse and selective DNx01 sparse driver line.

7) – It seems that it was possible to identify the DNx01s due to their strong sensory responses and large axons that were easily distinguishable in EM stacks and functional imaging. It would be nice if the authors could discuss a little further whether and how it will be feasible to expand this approach to other DNs in the future.

We agree with the Reviewer that the identification of DNx01s was facilitated by the unique properties of these neurons. We believe that additional tools may be required to expand this approach to arbitrary DNs in the future. We now state in the revised manuscript:

“Our analysis of DNx01 illustrates a potential road map for combining functional, topological, and morphological data to uncover the identity of individual DNs from population recordings. Notably, our efforts were facilitated by DNx01's unusual bilaterally asymmetric functional properties and large caliber axons. Taking this route to identify another, arbitrary DN would likely be more challenging and may require additional tools. For example, after functional imaging of DN populations, one might focally photoactivate GFP within individual DN axons of interest (Datta et al., 2008; Ruta et al., 2010). Subsequent morphological imaging could then be compared with DNs reconstructed in connectomics datasets (Zheng et al., 2018; Phelps et al., 2021).”

8) In l. 254, the authors suggest that turning might be driven by asymmetries in VNC networks rather than by asymmetric activation of VNC networks via DNs. This model is hard to reconcile with existing knowledge about motor control. It is possible for VNC networks to independently generate asymmetric activity of course (for example in response to unilateral local sensory inputs). However, if DNs are not asymmetrically activated to drive turning, how would the brain be able to drive voluntary turns? What is the underlying model?

We agree with the Reviewer that this model is not well-supported by evidence regarding the asymmetric DN control of steering. For example, asymmetric DNa01 activity has been associated with steering (Chen, Hermans et al., *Nature Communications* 2018; Rayshubksy *bioRxiv*, 2021) and unilateral MDN stimulation yield backward turning (Sen et al., *Current Biology* 2017). Therefore, we have removed this model from our revised manuscript.

We now state: “Simple models for locomotor control (Braitenberg, 1986) suggest that turning can be controlled by the relative activities of DNs on one side of the brain versus the other. This is supported by studies showing that flies generate turning during asymmetric activation or asymmetric activity of DNa01 neurons (Chen, Hermans et al., 2018; Rayshubskiy et al., 2020) and MDNs (Sen et al., 2017). To examine the degree to which this spatial asymmetry extends beyond pairs of neurons to much larger DN populations, we quantified the spatial location of turn-encoding DNs in the cervical connective.”

9) The authors use NeuroMechFly, a biomechanical simulation of the *Drosophila* body, to play back movements recorded by their motion capturing pipeline and detect collisions between the front legs and the right and left antennae. What is the reason for using such an indirect approach to detect potential antennal deflections? It seems the authors should be able to detect antennal deflections unambiguously in their video recordings. From looking at the supplemental videos, the leg movements of the model and the fly do not always seem to match perfectly (as would be expected from an approximation). Did the authors verify that their predictions were accurate?

We used NeuroMechFly to quantify collisions between the front legs and antennae because it provides a relatively rapid way to quantify contacts and, most importantly, because it is surprisingly difficult to confidently manually detect collisions in our low contrast camera images. This is because the cameras were intentionally focused on the legs to facilitate DeepFly3D joint pose estimation and the focal depth is not sufficient to resolve both the antennae and legs at sufficiently high contrast. We selected the data for Video 5 as the most clear example of individual legs contacting individual antennae. We agree with the Reviewer that future work focusing on leg-antennal contacts should explicitly include an in-depth manual validation like that performed to verify leg-ground contacts in the manuscript for NeuroMechFly (Lobato-Ríos et al., 2022, Extended Data Figure 7).

10) The definition of 'posterior movement' (l.127) is vague. Does this include every instance of abdominal bending? Were hind leg movements and abdominal bending treated the same way in the analysis? Why would that be a reasonable simplification? It would be nice if the authors could expand on this a little bit.

Classified behaviors were quantified using leg joint angles. Thus, ‘posterior movements’ do not include every instance of abdominal bending. Initially, our behavior classification included separate classes for hindleg grooming and abdominal grooming. However, these behaviors were rare and highly overlapping in our confusion matrix. Therefore, we grouped them together. We now expand on the definition of ‘posterior movements’ in the revised manuscript. We state:

“…and posterior movements (a grouping of rare hindleg and abdominal grooming movements)”.

11) The Discussion of the implications should be expanded, in particular, to include parallels to the ascending neurons, whose activity also seems to correlate with higher order/larger scale representations.

In the Discussion of the revised manuscript we now state: “Interestingly, many ascending neurons (AN) have also been shown to encode walking (Chen et al., *bioRxiv* 2022; Fujiwara et al., *Neuron* 2022) with a large fraction projecting to the gnathal ganglia (GNG), a brain region that is also heavily innervated by DNs (Namiki et al., *eLife* 2018). Thus, we speculate that ANs and DNs may be directly connected---possibly to mediate action selection (Mann et al., *Neuron* 2013; Bidaye et al., *Science* 2014)---and that this may lead to similar functional encoding.”

An explicit comparison to electrophysiological recordings of DNs should be included. An advantage of calcium imaging over electrophysiological recordings is the population aspect – signal can be compared among neurons to determine patterns and co-activation. How was this employed here?

In the Discussion of the revised manuscript we now state:

“Although electrophysiology provides higher temporal resolution (e.g., measuring spike timing (Von reyn et al., *Nature Neuroscience* 2014)), neural population imaging serves an important complementary role in capturing the proportion and spatial locations of co-active neurons.”

Why express GCaMP in most DNs at once, rather than in specific DN split GAL4 lines? Whether fluorescence changes correlated with behaviors could have been explored in both cases.

In the Discussion of the revised manuscript we now state: “Electrophysiological recordings and calcium imaging of sparse DN driver lines can more easily enable links to be made between neural encoding and cellular identity. However, these approaches suffer from two major disadvantages. First, there exist split-Gal4 driver lines for only a small fraction of DNs (Namiki et al., *eLife* 2018). Second, determining the encoding of an equivalent number of neurons requires many more sparse neural recordings than population imaging experiments. Therefore, by focusing on population imaging our study allowed us to more rapidly survey the encoding of a large number of DNs.”